# Human immune and gut microbial parameters associated with inter-individual variations in COVID-19 mRNA vaccine-induced immunity

Masato Hirota[1,13], Miho Tamai[1,13], Sachie Yukawa[1,2,13], Naoyuki Taira[1,13], Melissa M. Matthews[3], Takeshi Toma[1], Yu Seto[1], Makiko Yoshida[1], Sakura Toguchi[1], Mio Miyagi[1], Tomoari Mori[4], Hiroaki Tomori[5], Osamu Tamai[6], Mitsuo Kina[7], Eishin Sakihara[8], Chiaki Yamashiro[9], Masatake Miyagi[10], Kentaro Tamaki[11], Matthias Wolf [3], Mary K. Collins[12], Hiroaki Kitano [2] & Hiroki Ishikawa [1✉]

COVID-19 mRNA vaccines induce protective adaptive immunity against SARS-CoV-2 in most individuals, but there is wide variation in levels of vaccine-induced antibody and T-cell responses. However, the mechanisms underlying this inter-individual variation remain unclear. Here, using a systems biology approach based on multi-omics analyses of human blood and stool samples, we identified several factors that are associated with COVID-19 vaccine-induced adaptive immune responses. BNT162b2-induced T cell response is positively associated with late monocyte responses and inversely associated with baseline mRNA expression of activation protein 1 (AP-1) transcription factors. Interestingly, the gut microbial fucose/rhamnose degradation pathway is positively correlated with mRNA expression of AP-1, as well as a gene encoding an enzyme producing prostaglandin E2 (PGE2), which promotes AP-1 expression, and inversely correlated with BNT162b2-induced T-cell responses. These results suggest that baseline AP-1 expression, which is affected by commensal microbial activity, is a negative correlate of BNT162b2-induced T-cell responses.

[1] Immune Signal Unit, Okinawa Institute of Science and Technology, Graduate University (OIST), Onna-son, Okinawa, Japan. [2] Integrated Open Systems Unit, OIST, Onna-son, Okinawa, Japan. [3] Molecular Cryo-Electron Microscopy Unit, OIST, Onna-son, Okinawa, Japan. [4] Research Support Division, Occupational Health and Safety, OIST, Onna-son, Okinawa, Japan. [5] Yaesu Clinic, Naha-city, Okinawa, Japan. [6] Akebono Clinic, Naha-city, Okinawa, Japan. [7] Kina Clinic, Naha-city, Okinawa, Japan. [8] Health Care Center of the Naha Medical Association, Naha-city, Okinawa, Japan. [9] Yamashiro Orthopedic Surgery Ophthalmology Clinic, Naha-city, Okinawa, Japan. [10] Arakawa Clinic, Naha-city, Okinawa, Japan. [11] Naha-Nishi Clinic, Department of Breast Surgery, Naha-city, Okinawa, Japan. [12] Research Support Division, Office of the Provost, OIST, Onna-son, Okinawa, Japan. [13] These authors contributed equally: Masato Hirota, Miho Tamai, Sachie Yukawa, Naoyuki Taira. ✉email: hiroki.ishikawa@oist.jp

Vaccines containing mRNA encoding SARS-CoV-2 spike antigen, such as Pfizer BNT162b2, can effectively protect people against COVID-19[1–6]. Innate immune sensing of BNT162b2 mRNA by cytosolic RNA sensors immediately after vaccination is required for subsequent activation of spike-specific T-cell and antibody responses[7]. A second dose of BNT162b is sufficient to induce detectable spike-specific antibody and T-cell responses in most individuals, but levels of adaptive immune responses vary widely among individuals[8,9]. Although inter-individual variation in BNT162b2-induced adaptive immunity is associated with several parameters, such as SARS-CoV-2 infection history, age, sex, and ethnicity[9–11], the cause of this variation remains largely unknown.

Recent studies focused on systems biological understanding of human vaccine responses provide important insight into factors associated with inter-individual variation in vaccine-induced adaptive immunity[12–14]. Immune states represented by the composition of immune cells and gene expression profiles in individuals are highly variable, plausibly due to genetic diversity and environmental factors such as gut microbial flora[15–17]. Through comprehensive analysis of immune states of blood cells at baseline and early vaccine responses, specific immune cell populations and transcripts have been identified as correlates of antibody or T-cell responses induced by vaccination against influenza virus, hepatitis B virus, and malaria[18–22]. Moreover, other studies reveal that gut microbes are also associated with vaccine-induced adaptive immunity[23–25]. Importantly, these factors can be predictors of vaccine responses and may be potential therapeutic targets to improve vaccine responses[26,27]. However, the variability of immune states and gut microbes that is associated with COVID-19 mRNA vaccine responses remains unclear. In this study, using a systems biology approach, we demonstrate that BNT162b2-induced human adaptive immune responses are associated with specific immune and gut microbial parameters.

## Results

**Study design and cohort characteristics for vaccine-induced adaptive immune responses.** In this study, we used a systems biology approach based on multi-omics analyses of human blood and stool samples. 96 healthy subjects participated in this study (Supplementary Fig. 1a), and data from 95 participants who received two doses of BNT162b2 at a 3- to 4-week interval were analyzed (data from one participant who was not able to receive the second dose in a timely manner due to severe side effects from the first dose were excluded from the analysis). BNT162b2 induces a remarkable increase in expression of genes related to innate immunity on 1 day after the first dose and 1–7 days after the second dose[28]. Accordingly, we collected blood samples at multiple time points (Fig. 1a) to analyze baseline responses (T1: before vaccination), innate immune responses (T2: day 2 ± 1 after the first dose; T3: day 2 ± 1 after the second dose; T4: day 8 ± 2 after the second dose), and long-term adaptive immune responses (T5: day 41 ± 3 after the second dose). In addition, to analyze the gut microbiome, we collected stool samples from all subjects once during the participation period (Fig. 1a).

To evaluate the level of vaccine-induced adaptive immunity, we measured the SARS-CoV-2 spike-specific antibody response in plasma and the T-cell response in peripheral blood mononuclear cells (PBMCs), by enzyme-linked immunosorbent assay (ELISA) and enzyme-linked immunospot (ELISpot) assay, respectively. As previously reported[29], we also detected an increase in spike-specific antibody and T-cell responses at T5 in all subjects, but with remarkable individual differences (Supplementary Fig. 2a, b). Nine subjects were seropositive for SARS-CoV-2 spike and nucleocapsid proteins at baseline (Supplementary Fig. 2c). To remove the effect of previous SARS-CoV-2 infection[30], we focused on 86 subjects who were seronegative for SARS-CoV-2 spike at baseline in subsequent analyses. Consistent with previous reports[31–33], we observed gender-associated differences in antibody and T-cell responses (Supplementary Fig. 2d, e) and an age-related decline of vaccine-induced antibody responses, but not T-cell responses (Supplementary Fig. 2f, g). There was no detectable correlation between vaccine-induced antibody and T-cell responses (Supplementary Fig. 2h). After vaccination, T cell responses against common cold human coronaviruses (HCoVs), which show high identity to SARS-CoV-2 in amino acid sequences of spike proteins, increased and were correlated with T-cell responses against SARS-CoV-2 (Supplementary Fig. 3a–c), indicating that BNT162b2 can induce cross-reactive T cells to HCoVs as reported[34,35].

We constructed profiles of immune cell populations and mRNA expression in PBMCs and profiles of gut microbiome using cytometry by time of flight (CyTOF), bulk RNA sequencing, and 16S ribosomal RNA gene sequencing analyses (Fig. 1a). To identify factors associated with vaccine-induced adaptive immune responses, we used two different approaches in data analysis (Fig. 1b). In the first approach, 86 subjects seronegative for SARS-CoV-2 spike at baseline (hereafter referred to as the entire cohort) were randomly divided into discovery and validation cohorts (n = 43 each), and associations identified in the discovery cohort were subjected to be confirmed in the validation cohort (Supplementary Fig. 1b). In the second approach, data from the entire cohort (n = 86) were used to identify associations in order to increase statistical power. To remove the effect of age and sex, we performed partial correlation analyses. Through these analyses, we sought to identify immune cell populations, transcripts, and commensal microbial taxa and functions associated with vaccine-induced antibody and T-cell responses.

**Immune cell populations associated with BNT162b2-induced adaptive immune responses.** Using CyTOF data of the discovery cohort, we analyzed correlations between the frequency of 17 major immune cell populations (Supplementary Fig. 4a) and vaccine-induced adaptive immunity (antibody or T-cell responses at T5). This revealed that the frequency of monocytes at T5 was higher in high-T-cell responders than in low-T-cell responders (Supplementary Fig. 4b), which was confirmed in analysis of the validation cohort (Fig. 2a). In analysis of the entire cohort, we found that T-cell responses were not only positively correlated with the frequency of monocytes, but also positively and inversely correlated with the frequency of regulatory T (Treg) cells and several T cell subsets, respectively (Fig. 2b and Supplementary Fig. 4c, d). Time course analysis using data of high- and low-T-cell responders (top 20 and bottom 20 subjects in T-cell responses at T5) in the entire cohort showed vaccine-induced increases and decreases in the frequency of monocytes in high-T-cell responders (only at T5) and in low-T-cell responders (from T2 to T5), respectively (Fig. 2c and Supplementary Fig. 4e). Thus, the frequency of monocytes, which changes in the vaccine response, is a positive correlate of vaccine-induced T-cell responses.

**Transcripts associated with BNT162b2-induced adaptive immune responses.** To identify genes and pathways associated with BNT162b2-induced adaptive immunity, we next analyzed bulk RNA-seq data of PBMCs at T1 and T4. Of the 86 subjects who were seronegative for SARS-CoV-2 spike at baseline, sequence data from 80 (at T1) and 78 (at T4) subjects passed quality control. We observed that vaccination altered expression of 2296 genes, including genes related to plasma cells and B cells (Supplementary Fig. 5a, b). Using the discovery cohort, we

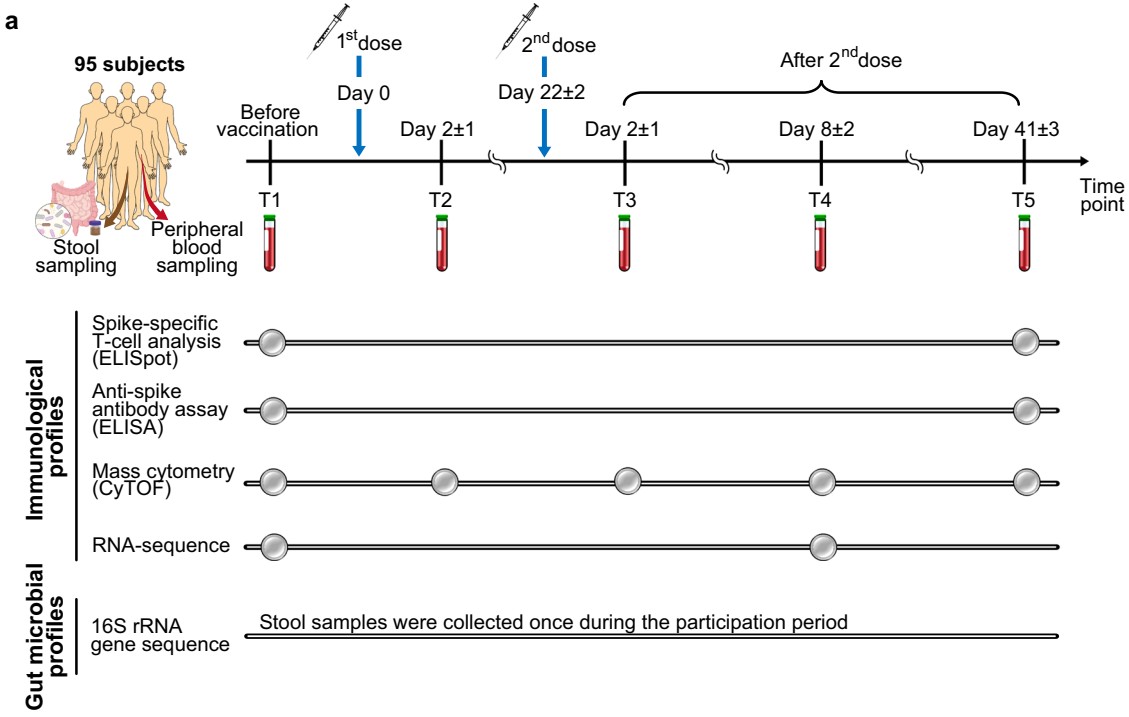

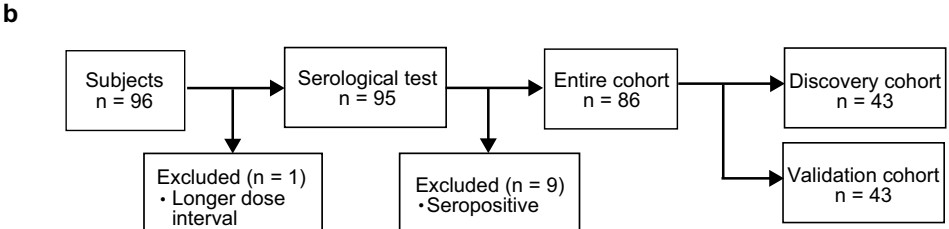

**Fig. 1 Study design. a** Schematic diagram showing blood and stool sample collection and analysis performed in this study. Samples from 95 subjects who received two doses of BNT162b2 at 3–4-week intervals were analyzed. **b** Schematic diagram showing data analysis approaches. The entire cohort (86 subjects who were seronegative for SARS-CoV-2 spike at baseline) were divided into the discovery and validation cohorts ($n = 43$ each). The discovery cohort or entire cohort was used to identify factors associated with COVID-19 antibody or T-cell responses, and factors identified in the discovery cohort were subjected to confirmation in the validation cohort.

performed gene set enrichment analysis (GSEA) to identify biological processes associated with vaccine-induced adaptive immunity. This revealed that a blood transcriptional module (BTM)[36] related to the activation protein 1 (AP-1) transcription network at T1, but not at T4, was negatively associated with T-cell responses (Supplementary Fig. 5c), which was confirmed in analysis of the validation cohort (Fig. 3a). Additionally, GSEA using the entire cohort data showed that a BTM related to the AP-1 transcription network at T1 was positively and negatively associated with antibody responses and T-cell responses, respectively (Fig. 3b).

One gene (at T4) and 130 genes (53 genes at T1, 77 genes at T4) were differentially expressed (log2 FC > 0.5, adjusted $P < 0.05$) in high- vs low-antibody responders and in high- vs low-T-cell responders in the entire cohort, respectively (Fig. 3c and Supplementary Fig. 5d). Notably, AP-1 transcription factors, such as *FOS, FOSB*, and *JUN* were highly expressed in low-T-cell responders at T1, but not at T4 (Fig. 3c). Gene regulatory network analysis of differentially expressed genes (DEGs) between high- and low-T-cell responders identified *FOS, JUN*, and *monocyte enhancer factor 2D* (*MEF2D*), which were highly expressed in low-T-cell

responders, as potential regulators for many DEGs (Fig. 3d). Baseline expression of *FOS* and *MEF2D*, but not *JUN*, was inversely correlated with vaccine-induced T-cell responses (Fig. 3e). Since AP-1 pathway was associated with T cell responses in GSEA, we assessed whether this is the case for other AP-1 family genes and found that expression of *activating transcription factor 3* (*ATF3*) and *FOSB* was also inversely correlated with T-cell responses (Fig. 3f). An additional RNA-seq analysis of high and low T-cell responders ($n = 19$ each) in the entire cohort at T5 showed that expression of *FOSB, JUND*, and *FOS like 2* (*FOSL2*), but not *FOS* and *ATF3*, was significantly higher in low-T-cell responders (Supplementary Fig. 5e). Consistent with the correlation between the monocyte frequency and vaccine-induced T cell responses at T5 (Fig. 2a, b), a BTM related to monocyte biology was positively associated with T-cell responses (Supplementary Fig. 5f). Taken together, we identified baseline expression of a subset of AP-1 genes *FOS, FOSB*, and *ATF3* as negative correlates of vaccine-induced T-cell responses.

**Baseline *FOS* expression is associated with ex vivo responses of PBMCs to BNT162b2 mRNA.** Next, we sought to investigate

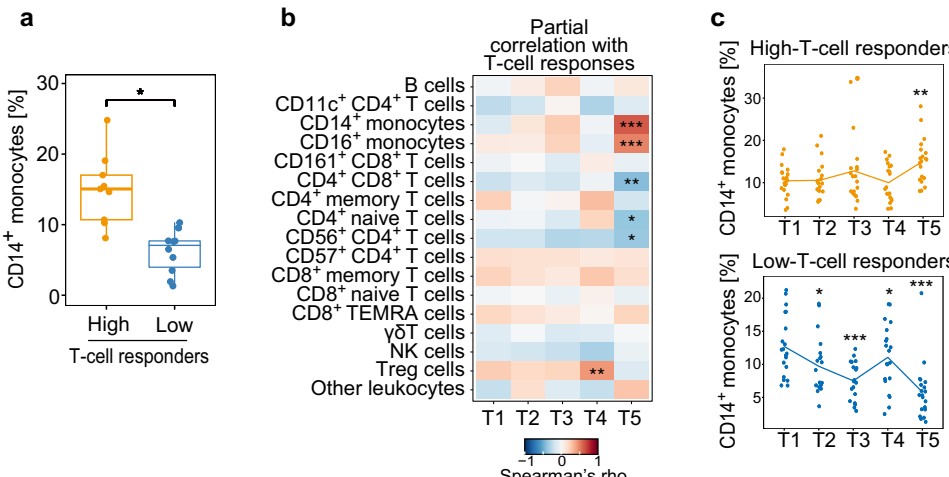

**Fig. 2 Late monocyte responses are associated with BNT162b2-induced adaptive immunity.** Frequency of immune cell populations in PBMCs was analyzed by CyTOF. **a** Frequency of CD14$^+$ monocytes in high- and low-T-cell responders ($n = 9$ and 10, respectively) in the validation group. p values were calculated with Wilcoxon signed rank tests with Benjamini–Hochberg FDR correction (*$P < 0.05$). Boxes show median and 25th–75th percentiles, and whiskers show the range. **b** Heat map showing the correlation between the frequency of immune cell populations and vaccine-induced T-cell responses in the entire cohort ($n = 86$). TEMRA, terminally differentiated effector memory; NK, natural killer. Partial correlation analyses with adjustments for age and sex were performed with Spearman's correlation tests with Benjamini–Hochberg FDR correction (*$P < 0.05$, **$P < 0.01$, ***$P < 0.001$). **c** Kinetics of the frequency of CD14$^+$ monocytes in PBMCs during vaccine response. High-T-cell responders (upper panel, $n = 20$) and low-T-cell responders (lower panel, $n = 20$) in the entire cohort were analyzed. $P$ values were calculated with the Wilcoxon signed rank tests with Benjamini–Hochberg FDR correction (*$P < 0.05$, **$P < 0.01$, ***$P < 0.001$).

whether transcriptomic signatures related to innate immune responses are associated with BNT162b2-induced adaptive immunity. To this end, we performed bulk RNA-seq analysis of PBMCs stimulated with BNT162b2 mRNA for 6 h ex vivo, because relatively large time lags in our blood sampling did not likely allow us to evaluate dynamic gene expression in BNT162b2-induced innate immunity. BNT162b2 mRNA stimulation upregulated genes related to type I interferon (IFN) responses (Supplementary Fig. 6a, b). GSEA in the entire cohort revealed that a BTM related to type I IFN responses was negatively and positively associated with antibody responses (Fig. 4a) and T-cell responses (Fig. 4b), respectively. Consistent with this, *interferon beta 1* (*IFNB1*) expression was correlated with vaccine-induced T cell responses (Fig. 4c) and was significantly higher in high-T-cell responders than low responders, which was confirmed by qPCR analysis (Fig. 4d). Moreover, *IFNB1* expression was inversely correlated with baseline expression of *FOS*, *FOSB*, and *JUN* (Fig. 4e). Analysis of PBMCs stimulated with BNT162b2 mRNA encapsulated with lipid nanoparticles also showed higher *IFNB1* expression in high-T-cell responders (Supplementary Fig. 6c).

To further investigate how baseline expression of AP-1 transcription factors is associated with early vaccine response, we performed single-cell RNA-seq (scRNA-seq) analysis of PBMCs of subjects who exhibited high or low *FOS* expression in the bulk RNA-seq analysis (high- and low-*FOS* subjects, $n = 4$ each) in the absence or presence of ex vivo stimulation with BNT162b2 mRNA. This experimental setting allowed us to evaluate the association between expression of *FOS* and other genes at baseline and in early innate immune response (6 and 16 h after BNT162b2 mRNA stimulation) in specific cell populations (Fig. 5a). Unsupervised clustering identified nine major immune cell populations whose frequencies were comparable between high- and low-*FOS* subjects (Supplementary Fig. 7a). BNT162b2 mRNA stimulation upregulated genes related to RIG-I-like receptor signaling and type-I IFN response, particularly in the monocyte population (Supplementary Fig. 7b).

We found that *FOS* was expressed all over the immune cell populations that we detected in unsupervised clustering analysis,

with the highest expression in CD14$^+$ monocytes, in the absence of BNT162b2 mRNA stimulation (Fig. 5b). As expected, *FOS* expression was significantly higher in high-*FOS* than low-*FOS* subjects (Fig. 5b). However, *FOS* expression was reduced in response to BNT162b2 mRNA stimulation in most PBMC subpopulations (Fig. 5b and Supplementary Fig. 7c). To investigate genes associated with baseline *FOS* expression in each cluster, we next performed GSEA on a ranked gene list based on changes in expression between high- and low-*FOS* subjects. This showed that GO terms related to baseline immunity, such as chemotaxis in CD14$^+$ monocytes, the tumor necrosis factor (TNF) signaling pathway in CD4$^+$ T cells, and the Toll-like receptor signaling pathway in CD8$^+$ T cells, were associated with high-*FOS* subjects at baseline (Fig. 5c and Supplementary Fig. 7d). In contrast, upon BNT162b2 mRNA stimulation, GO terms related to T cell activation, such as response to IFN-γ in CD4$^+$ T cells and responses to virus in CD8$^+$ T cells, were associated with low-*FOS* subjects (Fig. 5c and Supplementary Fig. 7d). Taken together, these results indicate that *FOS* expression is positively associated with expression of genes related to baseline immune cell activity, but it is negatively associated with that related to type I IFN responses and T cell activation upon BNT162b2 mRNA stimulation ex vivo.

**Gut microbes associated with BNT162b2-induced adaptive immune responses.** To assess the association between commensal gut microbes and vaccine-induced adaptive immunity, we next analyzed 16 S ribosomal RNA gene sequencing data. There was no difference in Shannon's diversity index in high- vs low-antibody responders and in high- vs low-T-cell responders in the entire cohort (Supplementary Fig. 8a). Linear discriminant analysis effect size (LEfSe) analysis in the discovery cohort identified several taxa associated with antibody or T-cell responses, but none of them were confirmed in the validation cohort. However, LEfSe analysis in the entire cohort identified 23 taxa and 11 taxa that were differentially enriched in high- vs low-antibody

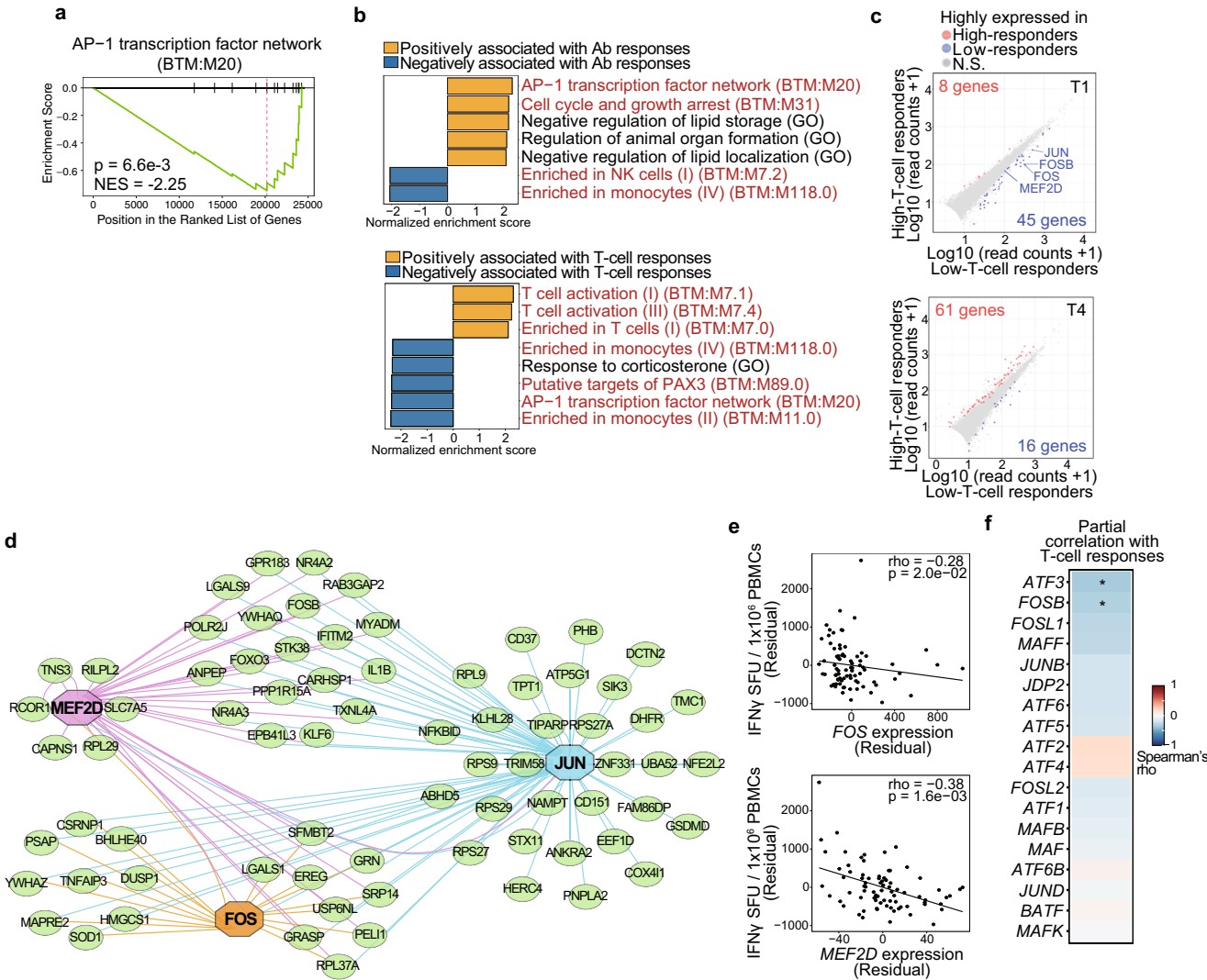

**Fig. 3 Transcripts associated with BNT162b2-induced adaptive immunity.** Transcriptomes of PBMCs isolated at time points T1 and T4 were analyzed by bulk RNA-seq. **a** AP-1 transcription factor network was identified in GSEA on a ranked gene list based on the Spearman's correlation coefficient between RNA expression and vaccine-induced T-cell responses in the validation cohort ($n = 43$). NES normalized enrichment score. **b** GSEA on a ranked gene list based on the Spearman's correlation coefficient between RNA expression and vaccine-induced T-cell or antibody responses in the entire cohort ($n = 86$). Immune-related pathways are shown in red. **c** Scatterplots showing DEGs between high- ($n = 18$ at T1, $n = 19$ at T4) and low- ($n = 19$ at T1, $n = 18$ at T4) T-cell responders in the entire cohort. DEGs: differentially expressed genes (log2 FC > 0.5, adjusted $P < 0.05$). Blue and red dots indicate genes that were highly expressed in the sample groups shown on the X axis and Y axis, respectively. N.S. not significant. **d** Gene regulatory network analysis of DEGs between high- and low-T-cell responders in the entire cohort. **e** Scatterplots showing correlations between vaccine-induced T-cell responses and expression of FOS and MEF2D. SFU, spot-forming units. Spearman's rho coefficient and P values are indicated in the plots. **f** Heat map showing correlations between vaccine-induced T-cell responses and expression of AP-1 genes ($n = 86$). **a, b, e, f** Partial correlation analyses with adjustments for age and sex were performed with Spearman's correlation tests with Benjamini–Hochberg FDR correction.

responders and in high- vs low-T-cell responders, respectively (Fig. 6a, b). There were no significant correlations between these taxa and vaccine-induced antibody or T cell responses in analysis with adjustments for age, sex, and stool sampling timing (Supplementary Fig. 8b, c).

We next searched for functions of gut microbiota that are associated with vaccine-induced adaptive immunity using a metagenome prediction tool, phylogenetic investigation of communities by reconstruction of unobserved states (PICRUSt2). In the analysis of the entire cohort, but not the discovery cohort, we found that the fucose/rhamnose degradation pathway of gut microbiota was inversely correlated with vaccine-induced T-cell responses (Supplementary Fig. 9a). Partial correlation analysis confirmed that the correlation between the fucose/rhamnose

degradation pathway and T-cell responses was independent of age, sex, and fecal sampling timing (Fig. 6c). The fucose/rhamnose degradation pathway converts fucose to lactaldehyde, which in turn is converted to (S)-1,2-propanediol or pyruvate (Fig. 6d). Among enzymes involved in this pathway, abundances of genes encoding L-fucose mutarotase and L-fuculokinase were significantly higher in microbiomes of low-T-cell responders (Fig. 6e and Supplementary Fig. 9b). Furthermore, we found that *Blautia*, which was enriched in low-T-cell responders (Fig. 6b), was a dominant taxon that encodes L-fucose mutarotase (Supplementary Fig. 9c, d).

Analysis of recently reported metagenome sequence data on gut microbiota before and after COVID-19 vaccination[25] showed that there was no significant change in the abundance of genes

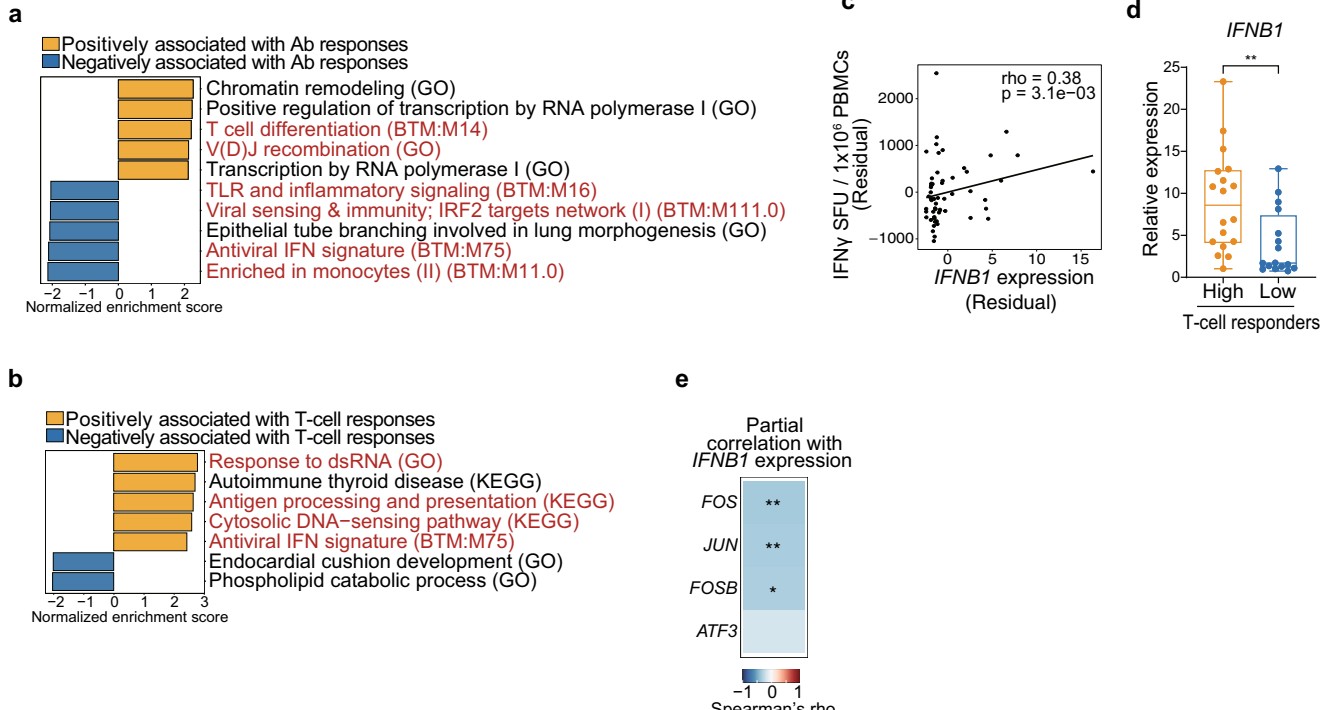

**Fig. 4 Association between baseline *FOS* expression and ex vivo type I IFN responses induced by BNT162b2 mRNA.** PBMCs collected from subjects (*n* = 86) in the entire cohort at T1 were stimulated with BTN162b2 mRNA for 6 h and analyzed by bulk RNA-seq (**a–c**, **e**) or qPCR (**d**). **a**, **b** GSEA on a ranked gene list based on Spearman's correlation coefficient between RNA expression and vaccine-induced antibody responses (**a**) or T-cell responses (**b**). Immune-related pathways are shown in red. **c** Correlation analysis between *IFNB1* expression induced by ex vivo BNT162b2 mRNA stimulation and vaccine-induced T cell responses. **d** *IFNB1* mRNA expression in high- (*n* = 18) and low- (*n* = 16) T-cell responders in the entire cohort was analyzed by qPCR. The *P* value was calculated with Wilcoxon signed rank test (**P < 0.01). Boxes show median and 25th–75th percentiles, and whiskers show the range.
**e** Correlation analysis between *IFNB1* expression induced by ex vivo BNT162b2 mRNA stimulation and baseline expression of *FOS*, *FOSB*, *JUN*, or *ATF3*.
**c**, **e** Partial correlation analyses with adjustments for age and sex were performed with Spearman's correlation tests with Benjamini–Hochberg FDR correction.

related to fucose/rhamnose degradation 1 month after vaccination (Supplementary Fig. 9e), suggesting that the activity of this metabolic pathway is largely not influenced by COVID-19 mRNA vaccination. Consistent with this, there was no detectable correlation between stool sampling timing and fucose/rhamnose degradation (Supplementary Fig. 9f). To evaluate the gut microbial fucose degradation activity, we anaerobically incubated stool slurry in the presence of fucose in vitro for 20 h and measured fucose levels in the culture media. The results confirmed that samples predicted by PICRUSt2 to have high fucose-rhamnose degrading activity reduced fucose more than those predicted to have low activity (Fig. 6f). Taken together, these data indicate that the gut microbial fucose/rhamnose degradation pathway is a negative correlate of vaccine-induced T-cell responses.

**The gut microbial fucose/rhamnose degradation pathway is associated with AP-1 expression.** Finally, we investigated whether the gut microbial fucose/rhamnose degradation pathway is associated with baseline expression of transcription factors that we identified as correlates of vaccine-induced T-cell responses. This showed that the gut microbial fucose/rhamnose degradation pathway was positively correlated with baseline *FOS*, *FOSB*, and *ATF3* expression in PBMCs (Fig. 7a and Supplementary Fig. 10a–c). Fucose/rhamnose degradation generates (*S*)-1,2-propanediol and pyruvate, which in turn leads to generation of short-chain fatty acids (SCFAs) (Fig. 7b). SCFAs derived from intestinal bacteria modulate host immune responses by inducing

colonic Treg cell differentiation[37–39]. Furthermore, SCFAs induce production of prostaglandin E2 (PGE2), which upregulates AP-1 expression[40]. We found that abundance of genes related to fucose/rhamnose degradation was correlated with expression of *prostaglandin-endoperoxide synthase 2* (*PTGS2*, also known as *cyclooxygenase 2* [*COX2*]), which encodes an enzyme catalyzing production of PGE2, but not with the frequency of Tregs at T1(Fig. 7c and Supplementary Fig. 10d). *PTGS2* expression was also positively correlated with expression of AP-1 factors, *FOS*, *FOSB*, and *ATF3*, and inversely correlated with vaccine-induced T cell responses (Fig. 7d, e). These results support the hypothesis that gut microbial fucose/rhamnose degradation may upregulate *PTGS2*/PGE2 expression in PBMCs probably through SCFAs, thereby promoting AP-1 expression. Indeed, treatment with SCFAs, but not (*S*)-1,2-propanediol, significantly increased expression of *PTGS2* in PBMCs (Fig. 7f). Furthermore, PGE2 treatment enhanced expression of *FOS* in PBMCs (Fig. 7g). These results suggest a potential functional link from the gut microbial fucose/rhamnose degradation pathway to AP-1 gene expression in PBMCs.

## Discussion

In this study, we identified various human immune cell populations and transcripts as well as gut bacterial taxa and functional pathways that are associated with BNT162b2-induced vaccine responses, using a systems biology approach. Notably, the baseline transcription module related to the AP-1 transcription factor network was positively associated with BNT162b2-induced

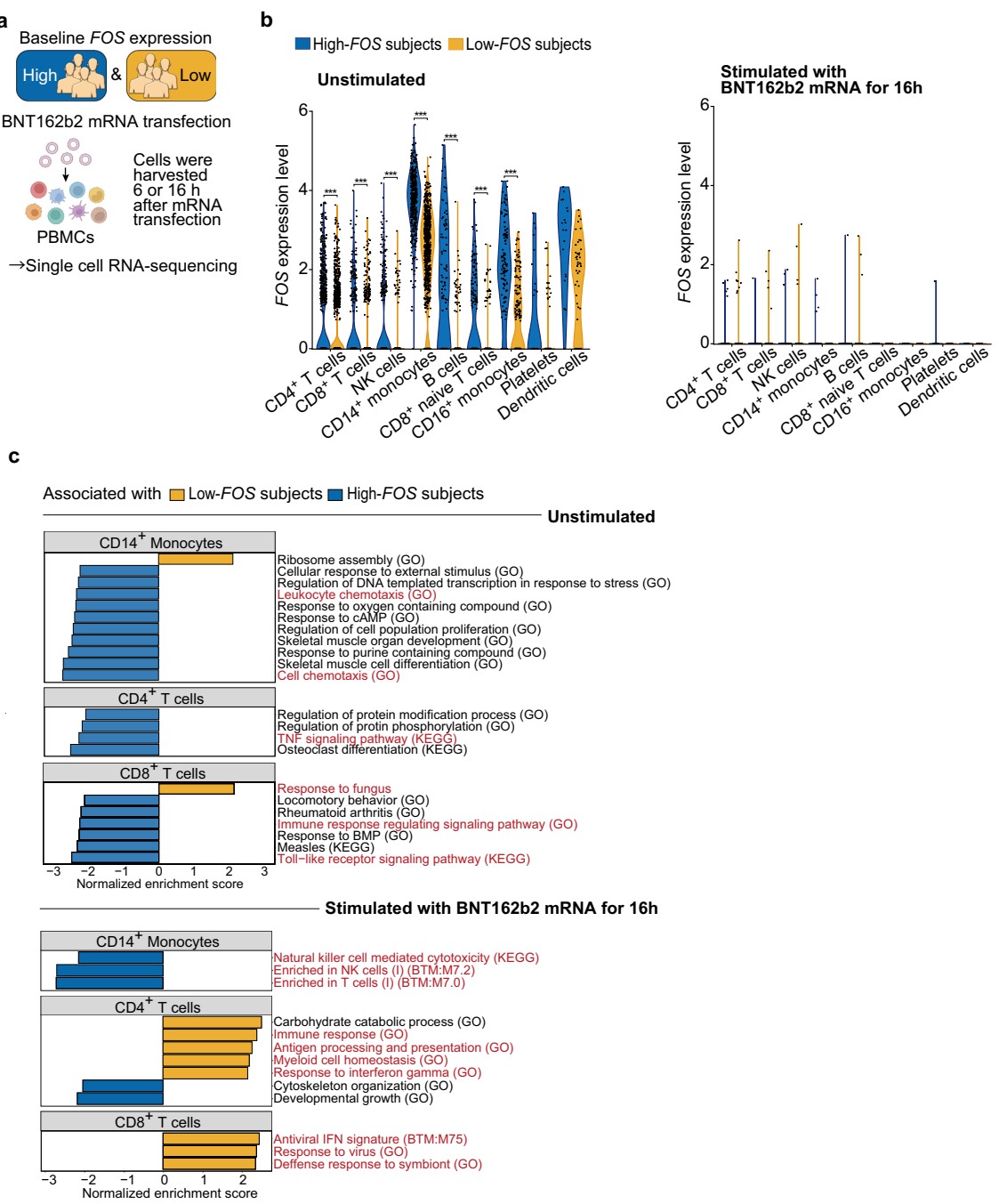

**Fig. 5 Association between baseline *FOS* expression and ex vivo responses of PBMC to BNT162b2 mRNA.** PBMCs isolated from subjects who exhibited high- or low-*FOS* expression in the bulk RNA-seq analysis (high- and low-*FOS* subjects, $n = 4$ each) were either unstimulated or stimulated with BNT162b2 mRNA for 6 or 16 h, followed by scRNA-seq analysis. **a** Schematic illustrating the experimental design of scRNA-seq of high- and low-*FOS* subjects. **b** Violin plots showing expression of *FOS* in PBMCs unstimulated (left panel) and stimulated with BNT162b2 mRNA for 16 h (right panel). *FOS* expression levels in each immune cell population were compared between high- and low-*FOS* subjects ($n = 4$ each). *P* values were calculated with Wilcoxon rank-sum tests with Benjamini–Hochberg FDR correction (***$P < 0.001$). **c** GSEA on a ranked gene list based on the change in expression in CD14+ monocytes, CD4+ T cells, and CD8+ T cells unstimulated or stimulated with BNT162b2 mRNA for 16 h between high- and low-FOS subjects. Immune-related pathways are shown in red.

antibody response and negatively associated with T-cell responses. Consistent with this, baseline expression of AP-1 genes (*FOS*, *FOSB*, and *ATF3*) was inversely correlated with T-cell responses. Furthermore, the gut microbial fucose/rhamnose degradation pathway was inversely correlated with T-cell responses. These findings advance our understanding of the contribution of immune and microbial factors to inter-individual variations in vaccine-induced adaptive immunity.

This study provides insight into the role of AP-1 genes in vaccine-induced T-cell responses. We observed that AP-1 expression in PBMCs rapidly decreased upon ex vivo stimulation with BNT162b2 mRNA, which is consistent with a recent report that expression of AP-1 genes such as *FOS* and *ATF3* was diminished in CD14+ monocytes by BNT162b2 vaccination[28]. Interestingly, the AS3-adjuvanted H5N1 pre-pandemic influenza vaccine also induces a decrease of AP-1 gene expression in

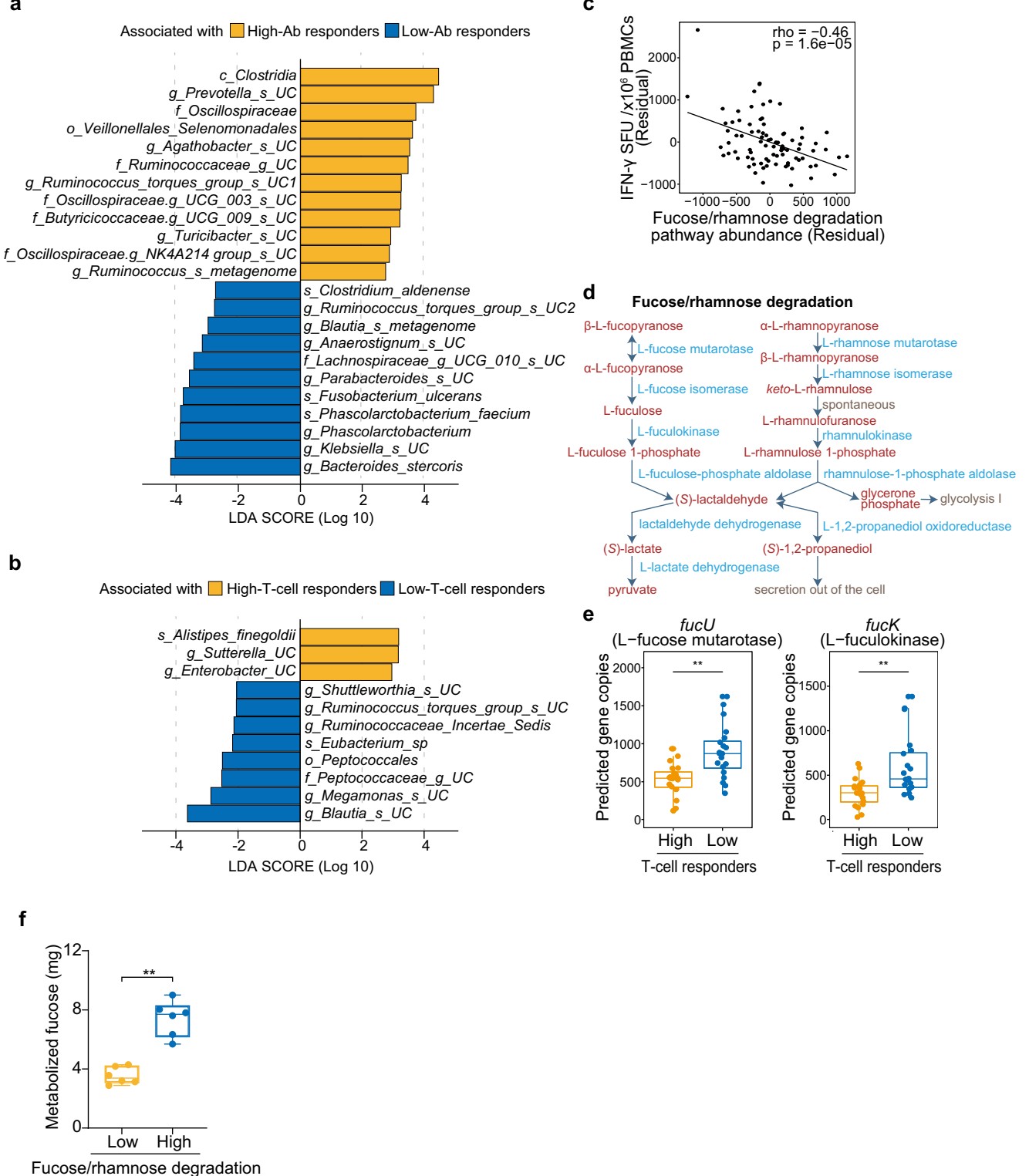

**Fig. 6 Gut microbes associated with BNT162b2-induced adaptive immunity.** Microbiomes of stool samples were analyzed by 16 S ribosomal RNA gene sequencing. **a**, **b** LEfSe analysis of gut microbes that were differentially abundant in high- vs low-antibody (Ab) responders (**a**) and in high- vs low-T-cell responders (**b**) in the entire cohort ($n = 20$ each). c, class; o, order; f, family; g, genus; s, species; UC, unclassified. **c** Scatterplot showing a correlation between the gut microbial fucose/rhamnose degradation pathway and vaccine-induced T-cell responses in the entire cohort ($n = 86$). Partial correlation analysis with adjustments for age, sex, and stool sampling timing was performed with Spearman's correlation tests. **d** Schematic showing the fucose/rhamnose degradation pathway. Metabolites and enzymes involved in the pathway are shown in red and blue, respectively. **e** Analysis of the abundance of predicted gene copies for ʟ-fucose mutarotase and ʟ-fuculokinase in high- and low-T-cell responders ($n = 20$ each) in the entire cohort. **f** Metabolized fucose levels in stool slurry incubated with fucose in vitro for 20 h. **e**, **f** P values were calculated with Wilcoxon rank-sum tests with Benjamini–Hochberg FDR correction (**P < 0.01). Boxes show median and 25th–75th percentiles, and whiskers show the range.

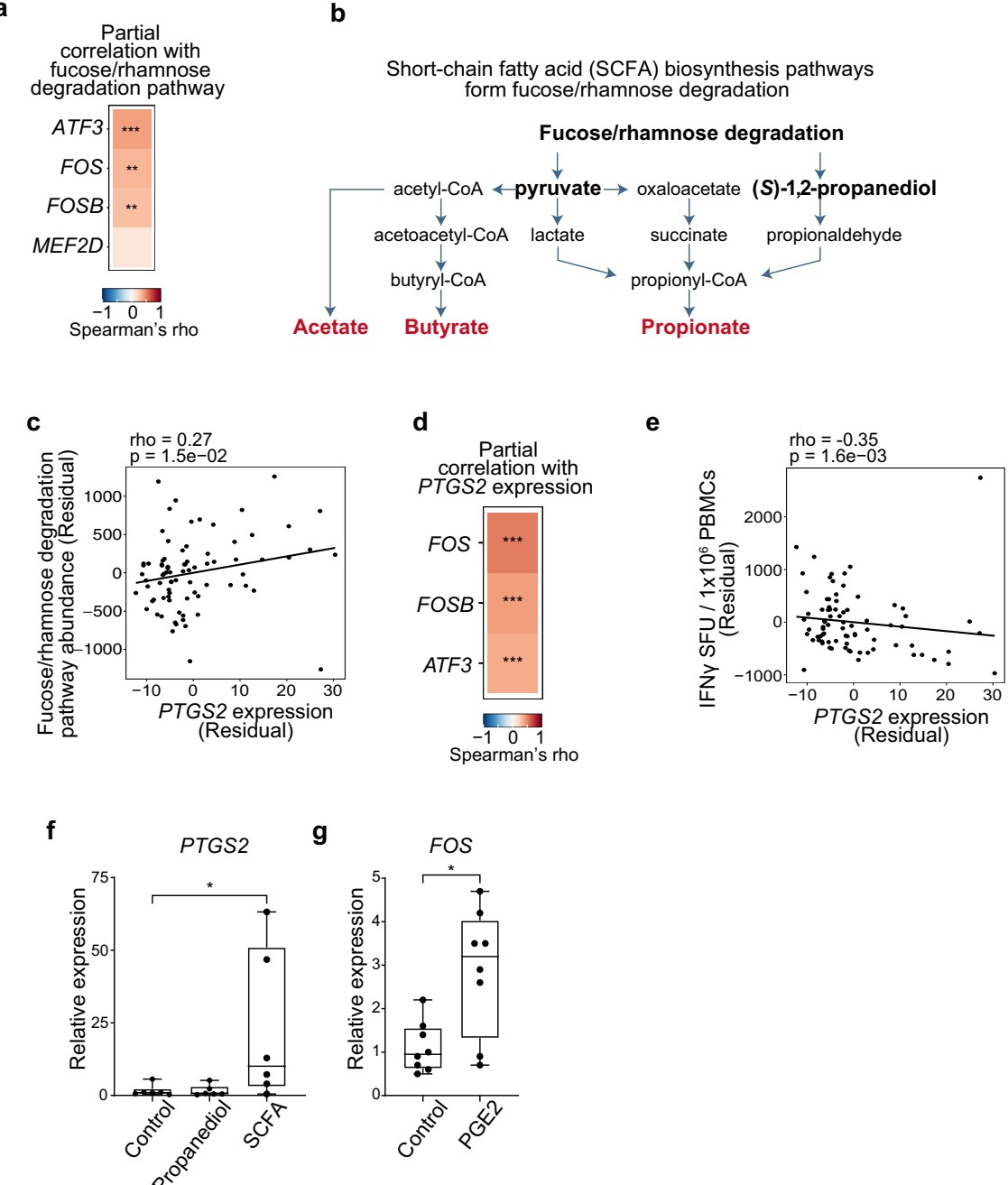

**Fig. 7 The gut microbial fucose/rhamnose degradation pathway is associated with AP-1 expression. a** Heat map showing the correlation between the gut microbial fucose/rhamnose degradation pathway and transcription factors associated with BNT162b2-induced T-cell responses in the entire cohort ($n = 86$). **b** Schematic showing the production of SCFAs from the fucose/rhamnose degradation pathway. SCFAs are shown in red. **c–e** Correlation analysis between *PTGS2* expression and fucose/rhamnose degradation (**c**), AP-1 expression (**d**), or vaccine-induced T-cell responses (**e**) in the entire cohort. **a**, **c–e** Partial correlation analyses with adjustments for age, sex, and stool sampling timing were performed with Spearman's correlation tests with Benjamini–Hochberg FDR correction (**$P < 0.01$, ***$P < 0.001$). **f** qPCR analysis of *PTGS2* mRNA levels in PBMCs untreated or treated with (S)-1,2-propanediol or SCFAs for 18 h ($n = 6$). *P* values were calculated using Friedman test followed by Dunn's multiple comparison test (*$P < 0.05$). **g** qPCR analysis of *FOS* mRNA levels in PBMCs untreated or treated with PGE2 for 18 h ($n = 8$). The *P* value was calculated with the Wilcoxon signed rank test (*$P < 0.05$). **f**, **g** Boxes show median and 25th–75th percentiles, and whiskers show the range.

monocytes through epigenetic silencing, which is maintained for at least 28 days[41]. Vaccine-induced alteration of AP-1 expression levels might explain why we detected association between vaccine-induced T-cell responses and expression of *FOS* and *ATF3* at baseline (T1), but not after vaccination (T4 or T5). AP-1 expression levels during vaccine response are associated with vaccine-induced cytokine expression[41], but how the difference in

baseline AP-1 expression affects vaccine response remains unknown. We found that *FOS* expression, which was inversely correlated with vaccine-induced T-cell responses, was positively associated with transcription modules related to baseline activity of CD14[+] monocytes and T cells. Furthermore, baseline *FOS* expression was negatively associated with transcription modules related to type I IFN responses and T cell activation upon

BNT162b2 mRNA stimulation ex vivo. These data suggest that baseline expression of *FOS* and other AP-1 factors in T cells and/or *FOS*-dependent control of baseline immune cell activity may inhibit type I IFN responses and T-cell activation induced by mRNA vaccines.

Our results suggest a functional link between the gut microbial fucose/rhamnose degradation pathway and the host immune system. We found that fucose/rhamnose degradation and AP-1 expression were positively correlated with *PTGS2* expression and inversely correlated with vaccine-induced T cell responses. The fucose/rhamnose degradation pathway can promote generation of SCFAs by several mechanisms, including cross-feeding of (S)-1,2-propanediol, a metabolic end-product of this pathway, between gut commensal bacteria resulting in production of propionate[42]. Our data suggest that SCFAs upregulate PGE2 production through upregulation of *PTGS2* expression, which in turn upregulates *FOS* expression in PBMCs. SCFAs promote mucosal Treg generation[37–39]. However, we did not observe an association between fucose/rhamnose degradation and the frequency of Tregs in PBMCs. Future studies will need to further explore the clinical significance and molecular mechanisms of interactions between the fucose/rhamnose pathway and vaccine-induced T-cell responses.

Our CyTOF analysis revealed a significant difference in the frequency of monocytes on Day 41 after the second dose between low- and high-T-cell responders. We observed a decrease of monocytes for at least two months after BNT162b2 vaccination in low-T-cell responders, but not in high responders. Conversely, there was an increase of monocytes between Days 8 and 41 after the second dose only in high-T-cell responders. These observations indicate remarkable heterogeneity in monocyte response induced by BNT162b2 vaccination. Infection and vaccination can affect monocyte development, homeostasis, and migration, thereby altering the frequency of monocytes in the blood[43,44]. Interestingly, vaccination with BCG, AS3-adjuvanted H5N1 pre-pandemic influenza (H5N1 + AS03) vaccine, or HIV vaccine induces innate memory monocytes that provide protection against non-related[41,45] and related viruses[46]. Epigenetic changes induced by H5N1 + AS03 are maintained in monocytes for at least 6 months, suggesting long-lasting trained immunity[41]. Furthermore, BCG vaccination enhances myelopoiesis, while decreasing lymphopoiesis in the bone marrow through persistent epigenetic modification of hematopoietic stem cells, which is plausibly important for innate immune memory of short-lived monocytes[47]. These epigenetic mechanisms in monocytes or hematopoietic stem cells might be involved in changes in the frequency of monocytes that we observed at T5 after BNT162b2-induced inflammatory responses.

This study successfully identified multiple correlates of BNT162b2-induced adaptive immune responses, but there are several limitations that should be noted. First, the associations identified only in the entire cohort analysis, such as the one related to gut microbial fucose/rhamnose degradation, have not been validated in other cohorts. Second, the relatively small sample size and the ethnic and geographic bias of participants in this study may have limited identification of correlates of adaptive immune responses. This may be one of the reasons why several enterobacterial taxa correlated with BNT162b2-induced antibody responses were identified in another study[25], but not in our study. Whether our findings are relevant to other ethnic groups is an important issue for future studies. Third, time lags in blood sampling over several days may have impeded identification of correlates of adaptive immunity that change dynamically in short time windows, such as genes induced by innate immunity. This issue was partly addressed by our RNA-seq analysis of PBMCs stimulated with BNT162b2 mRNA ex vivo, but the in vivo

relevance of findings from these analyses remains unknown. Fourth, we used the level of IFN-γ-secreting T cells as an indicator of T-cell responses for simple and accurate measurement by ELISpot assay, but analysis of CD4+ and CD8+ effector T cell subsets may be more informative. Fifth, high-throughput, scRNA-seq analysis and higher-resolution cell phenotyping by CyTOF will be required for a more comprehensive understanding of individual variation in vaccine-induced adaptive immunity.

In summary, we discovered several immune and microbial parameters at baseline and in the vaccine response that are associated with BNT162b2-induced antibody and T-cell responses, which provide insight into mechanisms of inter-individual variation in adaptive immunity. Our data suggest a key role of baseline AP-1 expression and the gut microbial fucose/rhamnose degradation pathway in inter-individual variation in mRNA vaccine-induced T-cell responses. Future studies should address the potential of these factors as baseline predictors of vaccine outcome and as therapeutic targets to improve vaccine responses.

## Methods

**Subjects**. The study was approved by the Okinawa Institute of Science and Technology, Graduate University (OIST) human subjects ethics committee (application HSR-2021–001). Ninety-six Japanese healthy volunteers (42 men and 53 women; average age, 52.4 ± 14.9 years; age range: 20–81 years) were recruited in Okinawa, Japan, between May 2021 and August 2021. All participants provided informed written consent. In total, 25 mL of peripheral blood was collected at each sampling. Stool samples were also collected from all participants once during the participation period. To identify parameters associated with COVID-19 vaccine responses, data from entire cohort (subjects seronegative for COVID-19 spike protein, $n = 86$) were analyzed. The entire cohort was randomly divided into the discovery and validation cohorts ($n = 43$ each) using the sample function in R software.

**PBMCs and plasma collection**. Using Leucosep tubes pre-filled with Ficoll-Paque Plus (Greiner; 163288), PBMCs and plasma were separated from blood samples collected in heparin-coated tubes (TERUMO; VP-H100K) as previously described[48]. Briefly, 25 mL of blood and 12 mL of AIM-V medium (Thermo; 12055091) were mixed, added to Leucosep tubes, and centrifuged at $1000 \times g$ at room temperature for 10 min. The upper yellowish plasma solution was collected and stored at –20 °C. The white layer containing PBMCs were washed twice with 22 mL of AIM-V medium by 7 min centrifugation, once at $600 \times g$ and a second time at $400 \times g$, and then cells were resuspended in 500 μL of CTL test medium (Cellular Technology Limited (CTL); CTLT-010). Fresh PBMCs were used for IFN-γ ELISpot assays, and the remaining PBMCs were stored in liquid nitrogen until use for other assays.

**SARS-CoV-2 antibody ELISA**. Anti-SARS-CoV-2 spike immunoglobulin G (IgG) ELISA assays were performed as previously described[49,50] with minor modifications. Briefly, 96-well plates were coated with 2-4 μg/mL HexaPro[51] spike protein overnight at 4 °C. The concentration was adjusted as necessary to optimize positive control signal reproducibility across protein purification batches. After blocking with 200 μL of PBST plus 3% milk, prepared serial dilutions of sera in PBST plus 1% milk were transferred to ELISA plates. Antibody incubation steps were carried out in an incubator at 20 °C. All other steps were carried out as described previously[49]. For data analysis, the endpoint titer was calculated using Prism 7 (GraphPad), and the background value was set at an OD492 of 0.2 arbitrary units (AU) based on previously reported data[52] and our data on a 10-specimen panel of negative controls. Accordingly, an ELISA reactivity greater than 0.2 AU was considered seropositive for SARS-CoV-2 spike.

Subjects seropositive for SARS-CoV-2 spike were subjected to SARS-CoV-2 N antibody test using anti-SARSCoV-2 N protein Human IgG ELISA Kit (ProteinTech) according to the manufacturer's instructions. Briefly, serum samples were diluted (1:100) and added to N-protein-coated plates for 30 min. After four washes, horseradish peroxidase-conjugated anti-human IgG antibodies were added and incubated for 30 min at room temperature. After four more washes, tetramethylbenzidine substrate was added and incubated for 10 min before adding the stop solution. Plates were read at 450 nm and 630 nm using SpectraMax iD3 (Molecular Devices).

**IFN-γ ELISpot assay**. Peptide pools for spike proteins from SARS-CoV-2 (JPT; PM-WCPV-S-1), HCoV-OC43 (GSC; PR30011), HCoV-NL63 (JER; PM-NL63-S-1), HCoV-229E (GSC; RP30010), and HCoV-HKU1 (JER; PM-HKU1-S-1) dissolved in dimethyl sulfoxide (DMSO) (500 μg/mL) were used for cell stimulation. Each peptide pool consisted of 15-mer peptides overlapping by 11 amino acids, covering the entire spike proteins from SARS-CoV-2 (UniProt ID: P0DTC2),

HCoV-229E (UniProt ID: P15423), HCoV-NL63 (UniProt ID: Q6Q1S2), HCoV-HKU1 (UniProt ID: Q5MQD0), and HCoV-OC43 (UniProt ID: P36334). IFN-γ ELISpot assays were performed using Human IFN-γ Single-Color Enzymatic ELISpot kits (CTL; hIFNgp-2 M), according to the manufacturer's instructions. Briefly, freshly isolated PBMCs ($2.5 \times 10^5$ cells per well) were stimulated with 1 mg/mL peptide solutions for 18 h. Negative controls (cells treated with equimolar amounts of DMSO) and positive controls (cells treated with 20 ng/mL phorbol 12-myristate 13-acetate (PMA) and 100 ng/mL ionomycin) were included in each analysis. After washing the plates, detection reagents included in the kits were added to wells. Spots were counted using a CTL ImmunoSpot S6 Analyzer (Cellular Technology Limited). To count antigen-specific spots, the number of background spots in a negative control well were subtracted from the number of spots in wells treated with peptide pools.

**CyTOF immunophenotyping.** Cryopreserved PBMCs were quickly thawed in a water bath at 37 °C and centrifuged for 5 min at 440×g. Cells were resuspended in TexMACS Medium (Miltenyi Biotec), treated with DNase I (100 U/mL) in the presence of 5 mM $MgCl_2$ for 15 min, centrifuged and resuspended in staining buffer, followed by barcoding with different combinations of Maxpar human anti-CD45 antibodies labeled with 106 Cd, 110 Cd, 111 Cd, 112 Cd, 113 Cd, or 114 Cd (Fluidigm). 18-20 barcoded PBMC samples were pooled ($1 \times 10^5$ cells/sample) and immunostained using a Maxpar Direct Immune Profiling Assay kit (Fluidigm, S00124) according to the manufacturer's protocol. This kit contained Live/dead intercalator-103Rh and the following 30 marker antibodies: CD45-89Y, CD196/CCR6-141Pr, CD123-143Nd, CD19-144Nd, CD4-145Nd, CD8a-146Nd, CD11c-147Sm, CD16-148Nd, CD45RO-149Sm, CD45RA-150Nd, CD161-151Eu, CD194/CCR4-152Sm, CD25-153Eu, CD27-154Sm, CD57-155Gd, CD183/CXCR3-156Gd, CD185/CXCR5-158Gd, CD28- 160Gd, CD38-161Dy, CD56/NCAM- 163Dy, TCRgd-164Dy, CD294-166Er, CD197/CCR7-167Er, CD14-168En, CD3-170Er, CD20-171Yb, CD66b-172Yb, HLA-DR-173Yb, IgD-174Yb, and CD127-176Yb. PBMC samples were washed three times with Cell Acquisition Solution (CAS) or CAS plus buffer (Fluidigm) and resuspended in the same buffer containing a 1/10 dilution of EQ beads (Fluidigm). Samples were analyzed (an average of $5 \times 10^4$ events/sample) with a Helios mass cytometer system (Fluidigm).

**CyTOF data analysis.** Flow cytometry standard (FCS) files were normalized using EQ beads and concatenated. Then files were de-barcoded using the barcode key file (Key_Cell-ID_20-Plex_Pd.csv) in the Fluidigm acquisition software (v. 6.7.1014). Clean-up gates for live single cells and elimination of non-cell signals were manually conducted using the web-plat software, Cytobank (v.9.1). To correct batch effects across CyTOF runs, signal intensities were normalized using cyCombine (v.0.1.8)[53]. Data were analyzed using a previously described R-based pipeline[54]. In brief, data were imported and transformed for analysis using the read.flowSet function from the flowCore (v.2.6.0) package[55] and the prepData with option (cofactor = 5) function from the CATALYST (v.3.16; https://github.com/HelenaLC/CATALYST) package, respectively. Clustering was based on the fastPG (v.0.0.8)[56] algorithm with default parameters. These clusters were visualized using t-distributed stochastic neighbor embedding (t-SNE) and subsequently annotated based on protein markers expression.

**Bulk-RNA seq.** Total RNA was isolated from PBMCs using Isospin cell and tissue RNA kit (Nippon Gene) or an RNAdvance v2 kit (Beckman Coulter) according to manufacturer instructions and quantified with an RNA HS Assay Kit (Thermo Fisher) and a Qubit Flex Fluorometer (Thermo Fisher). For transcriptome analysis, 10 μg of RNA were used for library preparation with a QuantSeq 3′ mRNA-Seq Library Prep Kit FWD for Illumina (Lexogen) according to the manufacturer's protocol for low-input RNA samples. To generate single-nucleotide polymorphism (SNP) calls for several donors whose samples were analyzed by scRNA-seq, cDNA libraries were prepared from 500 ng of RNA using a Collibri Stranded RNA Library prep Kit (Thermo Fisher) according to the manufacturer's protocol for degraded RNA samples. Libraries were quantified with a Qubit 1x dsDNA HS Assay Kit (Thermo Fisher) and a Qubit Flex Fluorometer (Thermo Fisher), and quality was assessed using D1000 ScreenTape and High Sensitivity D5000 ScreenTape with a Tapestation 2200 (Agilent). Pooled libraries were sequenced on a Novaseq 6000 instrument (Illumina) with $1 \times 100$-bp reads for transcriptome analysis and $2 \times 150$-bp reads for generation of SNP calls at the Sequencing Section at OIST.

**Bulk RNA-seq data processing.** To evaluate data quality, we applied FastQC (v.0.11.9) (www.bioinformatics.babraham.ac.uk/projects/fastqc/). Reads were further processed to remove adaptor and low-quality sequences using Trimmomatic[57] (v.0.39) software with the options (SLIDINGWINDOW:4:20 LEADING:20 TRAILING:20 MINLEN:20 HEADCROP:12). To align reads to the GRCh38 reference genome (Homo_sapiens.GRCh38.dna.primary_assembly.fa downloaded from Ensembl), we used HISAT2[58] (v.2.2). We counted the number of reads overlapping the genes in GENCODE (v.30) reference transcriptome annotations using featureCounts from Subread[59] (v.2.0.1) with flags (-s 1 -t gene). Samples with fewer than 300,000 total reads were excluded from the analysis. To detect differentially expressed genes between the high- and low- Ab or T-cell responders, we first filtered transcripts with an average read count of less than 5 and analyzed

statistical significance with the Wald test using DESeq2[60] (v.1.34.0). Gene set enrichment analysis based on blood transcriptional module (BTM), Kyoto Encyclopedia of Genes and Genomes (KEGG), and Gene Ontology collection (GO) was performed using the clusterProfiler[61] package (v.4.2.2). To predict regulators that explain the observed differential transcriptional program between the two groups, we used iRegulon[62] (v.1.3) through the Cytoscape (v.3.9.1) visualization tool. Analysis was performed on the putative regulatory region of 20 kb centered around the transcription start site using default settings.

**SNP calling.** Sequencing reads were adaptor- and quality-trimmed and then aligned to the human genome using the Hisat2 aligner. SNP calls were generated using a previously published protocol[63]. In brief, we used SAMtools[64] (v.1.12) to remove duplicates (command markdup). Then, we applied the BEDtools[65] (v.2.26.0) intersect to identify and remove SNPs in imprinted genes (http://www.geneimprint.org/ accessed: 3 January 2022) and SNPs in repeats (RepeatMasker annotation downloaded from the UCSC Genome Browser). Genotypes were obtained with SAMtools mpileup with options (-A -q 4 -t AD, DP) and BCFtools[66] (v.1.11-1) call (with options -m --O b -f GQ), using uniquely mapped reads. We used VCFtools[67] (v.0.1.16-2) to select SNPs with a depth ≥10 with options (-minDP 10) and a genotype quality ≥20 with options (-minGQ 20).

**Ex vivo PBMC stimulation with BNT162b2 mRNA.** The BNT162b2 cDNA sequence, including 5′ and 3′ untranslated regions[68], was synthesized by integrated DNA technology (IDT) and cloned into pCDNA3.1 (Thermo Fisher). Using PCR-amplified BNT162b2 cDNA with an upstream T7 promoter as a template, in vitro transcription of BNT162b2 mRNA was performed with a HiScribe T7 ARCA mRNA Kit with tailing (NEB) with 2.5 mM N1-Methylpseudouridine-5′-triphosphate nucleoside analog (TriLink BioTechnologies) instead of unmodified UTP. BNT162b2 mRNA was purified using a Monarch RNA cleanup kit (NEB) and dissolved in nuclease-free water.

PBMCs were seeded in a 96-well plate ($10^6$ cells/well), and stimulated by transfection with BNT162b2 mRNA (200 ng/well) using Lipofectamine MessengerMAX (Thermo Fisher) according to the manufacturer's instructions. Cells were harvested 6 or 16 h after mRNA transfection. In some experiments, PBMCs were stimulated with BNT162b2 mRNA encapsulated with lipid nanoparticles (LNP). To prepare LNP-BNT162b2 mRNA, BNT162b2 mRNA in citrate buffer (5 mM sodium citrate, 5 mM citric acid, and 150 mM sodium chloride, pH 4.5) was mixed with 2.3 mM ALC-0315 (Echelon Biosciences), 0.8 mM ALC-0159 (Echelon Biosciences), 0.5 mM DSPC (Echelon Biosciences), and 0.8 mM cholesterol (Echelon Biosciences) in ethanol using a syringe pump with LNPSC-SHMGHFL (NT Science). LNP-BNT162b2 mRNA was purified using an ultra-centrifuge filter (Merck Millipore, UFC 810024) and resuspended in PBS (20 ng/μL). PBMCs were seeded into a 96-well plate ($10^6$ cells/well) and stimulated with BNT162b2 mRNA/LNP (200 ng/well).

**scRNA-seq.** PBMCs unstimulated or stimulated with BNT162b2 mRNA for 6 h or 16 h were used for analysis. Cells from eight subjects were pooled in equal numbers and resuspended in ice-cold PBS with 0.04% BSA at a final concentration of 1000 cells/μL. Single-cell suspensions (about 20,000 cells) were then loaded on the 10X Genomics Chromium Controller (Supplementary Table 1). Libraries were generated using a Chromium Next GEM Single Cell 5′ v2 (Dual Index) Reagent Kit according to the manufacturer's instructions. A Quantitative PCR Bio-Rad T100 Thermal Cycler (Biorad) was used for a reverse transcription reaction. All libraries were quality controlled using a Tapestation (Agilent) and quantified using a Qubit Fluorometr (Thermo Fisher). Libraries were pooled and sequenced on an Illumina NovaSeq platform (Illumina) using the following sequencing parameters: read1-26-cycle, i7-10, i5-10, read2-90 with a sequencing target of 20,000 reads per cell RNA library.

**scRNA-seq data analysis.** The CellRanger Single-Cell Software Suite (v.6.0.0; 10x Genomics) was used to perform barcode processing and transcript counting after alignment to the GRCh38 reference genome with default parameters. To match single cells in the 10x RNAseq data to each donor and identify doublets, we used the software package demuxlet (v.2.0.1)[69], which uses variable SNPs between pooled individuals. To further analyze scRNA-seq data, we used the Seurat (v.4.1.0)[70] R package. Cells expressing >5% mitochondrial gene counts or expressing fewer than 500 genes were discarded using the subset function. Then, the NormalizeData and FindVariableFeatures functions were applied to each dataset before FindIntegrationAnchors, IntegrateData and ScaleData were called to combine and scale the data. Unsupervised clustering was applied in each dataset as follows: (i) The top variant genes selected by FindVariableFeatures were used as input for principal components analysis (PCA) to reflect major biological variation in the data. (ii) The top 15 principal components were used for t-SNE dimensional reduction with the RunTSNE function and unsupervised clustering. Specifically, the FindClusters function was used to cluster cells. (iii) After cell clusters were determined, marker genes for each cluster were identified by the FindAllMarkers function with default parameters. The AddModuleScore function was used to calculate the module score in each cell. Plots of expression of specific transcripts were created using the FeaturePlot function. To find differentially expressed genes

between high- and low-*FOS* groups, we used the FindMarkers function with the MAST algorithm[71]. Gene set enrichment analysis based on BTM, KEGG, and GO was performed using the clusterProfiler R package.

**16 S rRNA gene sequencing**. Fecal samples were stored at –80 °C until use. After thawing, approximately 50 mg of each stool sample was transferred into a 2-mL tube containing 0.1 mm zirconia/silica beads (BioSpec Products) and 3.0 mm zirconia beads (Biomedical Sciences). Stool samples were disrupted using a TissueLyser II (Qiagen) for 10 min at 30 Hz after adding the Inhibit EX buffer from the QIAamp Fast DNA Stool Mini Kit (Qiagen), and genomic DNA was extracted using the QIAamp Fast DNA Stool Mini Kit in accordance with the manufacturer's instructions. To amplify16S rRNA V3 and V4 regions, PCR was performed using Kapa Hifi Hotstart Ready Mix (KAPA Biosystems) with an amplicon PCR primer set (forward: 5′-TCG TCG GCA GCG TCA GAT GTG TAT AAG AGA CAG CCT ACG GGN GGC WGC AG-3′, reverse: 5′-GTC TCG TGG GCT CGG AGA TGT GTA TAA GAG ACA GGA CTA CHV GGG TAT CTA ATC C-3′). The PCR condition was: 95 °C for 3 min, followed by 25 cycles of 95 °C for 30 s, 55 °C for 30 s, and 72 °C for 30 s, and then 72 °C for 5 min. PCR products purified with AMpure XP beads (Beckman) were further amplified by another PCR using Kapa Hifi Hotstart Ready Mix with Nextra XT Index Primers from Nextra XT Index Kit (Illumina). PCR conditions were: 95 °C for 3 min, followed by eight cycles of 95 °C for 30 s, 55 °C for 30 s, and 72 °C for 30 s, and then 72 °C for 5 min. Library DNA was purified with AMpure XP beads and quantified using a Qubit 1x dsDNA HS Assay Kit. Samples were sequenced on an Illumina Miseq with 2x300bp reads at the Sequencing Section at OIST.

**16 S rRNA gene sequencing data analysis**. FASTQ files were analyzed using the QIIME2 pipeline[72] (QIIME2 version 2020.2). After conversion to the qza format, sequence data were demultiplexed and summarized using QIIME2 paired-end-demux. Then, sequences were trimmed and denoised with the dada2 plugin for QIIME2. Taxonomy was assigned using a naive Bayes-fitted classifier trained on the SILVA_132 reference database (SSURef_NR99_132_SILVA) with the feature-classifier plugin for QIIME2. The phylogenetic tree for diversity analysis was reconstructed using QIIME2 align-to-tree-mafft-fasttree. Diversity analysis was performed with QIIME2 core-metrics-phylogenetic. PICRUSt2 (v.2.4.2)[73] was used to determine predicted functions of bacterial communities. Comparisons of bacterial taxon abundance were performed with LEfSe Galaxy instance[74] using default parameters. In LEfSe analysis, reads assigned to the mitochondrial and chloroplast genomes were filtered out. In addition, taxa detected in less than 10% of participants ($n = 86$) or in less than 10% of a subset of participants ($n = 40$, top 20 and bottom 20 antibody or T cell responders) were excluded from the analysis.

**Analysis of fucose degradation activity using stool culture**. To evaluate fucose degradation activity, stool samples were cultured as previously reported with some modifications[75]. Briefly, fecal samples stored at –80 °C were thawed on ice and resuspended in phosphate buffer (0.1 M, pH 7.0) at 32% (wt/vol) to prepare the inoculum. Fermentation medium (14 mg/mL peptone, 0.3 mg/mL cysteine, 0.3 mg/mL sodium sulfide, and 1.2 µg/mL Resazurin) was supplemented with or without 50 mg/mL fucose (Sigma Aldrich), and was deoxygenated in a jar containing AnaeroPack as an oxygen absorber (A-03, Mitsubishi Gas Chemical), for 24 h before use. The inoculum (50 µL) and the fermentation medium (180 µL) were mixed in a 14 mL tube and incubated at 37 °C with shaking under anaerobic conditions in a jar containing AnaeroPack for 20 h. The reaction was stopped by incubating the culture tube on ice for 1 h, and fucose concentration was measured using L-FUCOSE Kit (Megazyme) in accordance with the manufacturer's instructions.

**Treatment of PBMCs with SCFA and PGE2**. PBMCs were seeded into a 96-well plate ($10^6$ cells/well), and treated with SCFAs (mixture of 0.6 mM acetate [Sigma-Aldrich], 0.2 mM propionate [Sigma-Aldrich], and 0.2 mM butyrate [Sigma-Aldrich]), 10 mM (*S*)-1, 2-Propanediol (Tokyo Chemical Industry), or 10 µM PGE2 (Nacalai tesque).

**RNA isolation and qPCR**. cDNA was synthesized using ReverTra Ace qPCR RT Kit (Toyobo) using 200 ng of total RNA in a 10-µL volume. cDNA samples were diluted fourfold by adding 30 µL sterile nuclease-free water and 10 µL of cDNA were used for PCR reactions. PCR was carried out using KAPA SYBR FAST qPCR Kit Master Mix (KAPA BIOSYSTEMS, KK4602) and primer sets (Supplementary Table 2) on a StepOnePlus Real-Time PCR System (Applied Biosystems).

**Statistics and reproducibility**. Statistical details for each experiment are included in the figure legends. Wilcoxon rank-sum tests and Wilcoxon signed-rank tests were performed using R (v.4.1.2) or GraphPad Prism (v.9.4.0). Correlation and partial correlation analyses were performed using Spearman's correlation tests in the stats R package (v.4.1.2). To remove the effects of age, gender, and fecal sampling timing in partial correlation analysis, we calculated Spearman's correlation coefficient using residuals from linear regression of ranked variables. P values were corrected using Benjamini–Hochberg false discovery rate (FDR) for multiple comparisons.

**Reporting summary**. Further information on research design is available in the Nature Portfolio Reporting Summary linked to this article.

## Data availability
Source data has been provided as Supplementary Data 1. Sequencing data that support the finding of this study have been deposited to DDBJ database under accession numbers PRJDB14078 (for bulk RNA-seq and 16S rRNA gene sequencing) and PRJDB14085 (for scRNA-seq). Metagenome sequence data on gut microbiota before and after COVID-19 vaccination is available in BioProject (PRJEB48269). Any other relevant data are available from the corresponding author upon request.

## Code availability
Source codes used for this study are available upon request.

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

## Acknowledgements

This study was in part supported by funding from COVID-19 AI and Simulation Project (Cabinet Secretariat) to HI, the Platform Project for Supporting Drug Discovery and Life Science Research (BINDS) from AMED under grant number JP18am010107 to M.W., and JST grant number JPMJPF2205 to H.K. We are also grateful to OIST Graduate University for its generous funding of the Immune Signal Unit. We thank physicians and nurses at KIN oncology Clinic and in Naha Medical association for excellent support to collect blood samples from donors and our laboratory members for valuable discussions. We also thank Steven D. Aird for editing the manuscript.

## Author contributions

M.H. performed data analysis of CyTOF, bulk and scRNA-seq, and 16 S rRNA gene sequencing. M.T. performed most of the experiments except for ELISA and scRNA-seq. S.Y. performed experiments and data analysis of ELISpot, CyTOF and 16 S rRNA gene sequencing. NT prepared the scRNA-seq library and did some other experiments. M. Matthews and M.W. carried out anti-SARS-CoV-2 spike IgG ELISA. T.T., Y.S., M.Y., S.T., and Mio M. collected PBMCs from blood samples. T.M., H.T., O.T., M.K., E.S., C.Y., Masataka M., and K.T. supervised volunteer recruitment and blood and stool sample collection. S.Y., M.C., H.K., and H.I. designed and supervised the study. M.H., M.T., and H.I. interpreted the data and wrote the manuscript.

## Competing interests

The authors declare no competing interests.

**Additional information**

