## [Peer Review File · Communications Biology]

Reviewers' comments:

Reviewer #1 (Remarks to the Author):

Hirota et al. employ a systems biological approach to determine correlates of inter-individuals variation observed in BNT162b2 vaccinees. Using a cohort of healthy volunteers (n=96 participants, of which 86 are seronegative for SARS-CoV-2 at baseline). Using CyTOF, they show that CD14+ monocyte frequency measured 42 days post second dose positively correlates with vaccine-induced T cell responses. Using transcriptomics analysis, they also show that an AP-1 transcription factor expression correlates negatively with T cell responses while a type I interferon response signature positively correlates with T cells. Furthermore, they show that Fucose/rhamnose degradation pathway, measured by metagenomic sequencing, correlates with AP-1 transcription factors thus negatively correlating with the T cell responses. While the objective of the study is of interest to the field, the study needs to be improved significantly.

1. A major issue is lack of validation. Since the primary goal of the study is to determine the correlates of variations in vaccine-induced responses, it is critical that the determinates are validated in an independent dataset. I realize they may not have an independent population, but they could randomly distribute the participants (~40 each) into two cohorts, and use one as discovery and the other as validation.
 2. How were the data normalized for age and sex Figures 3b, c and other datasets?
 3. It is intriguing that the CD14+ monocyte frequencies increase significantly at T5 (Fig. 3e, top panel), i.e., 42 days post second dose while it is relatively stable at the rest of the time points analyzed. What do the authors' think is the reason for such an increase at such a late time point?
 4. If the monocytes increase in frequency, bulk transcriptomics should also show an increase in monocyte-associated signatures increasing at the same time point (T5). Why did the authors not look at it? This could also be an additional layer of confirmation of the monocyte correlation with T cells?
 5. What do the authors posit as a biological explanation for a negative correlation between frequency of T cell subsets (Fig. 3c) and antigen-specific T cell response?
 6. Single-cell RNAseq of PBMCs is a nice way of validating bulk transcriptomics results, and provide further insights into the cellular origin of AP-1 genes. However, it appears that the authors analyzed only baseline PBMCs (stimulated in vitro with mRNA, see next point) and not T4, which is when the bulk transcriptomics analysis showed a correlation. Why is this? I am unclear if the single-cell data confirms AP-1 correlation with T cells.
 7. The in vitro system where they transfect in vitro transcribed mRNA is not comparable with in vivo administration of BNT162b2. The lipid nanoparticle used in BNT162b2 has enormous effects on the innate immune system (see Alameh et al. Immunity 2021). The in vitro mRNA transfection does not have this component.
 8. The gut microbiome data is interesting but it appears that the stool samples were not synchronized in collection time point. How is that divergence normalized?
- Overall, the goal of the study is interesting and important, but the manuscript needs significant improvement and importantly, provide validation.

Reviewer #2 (Remarks to the Author):

General

In this manuscript, Hirota et al describe multi-omics analysis of the individuals who received the vaccination of SARS-CoV-2. The analysis covers the ELISpot analysis for the T cells reactive to the spike protein, ELISA assays of the antibodies and CyTOF analysis of PBMC. The sampling was performed in a time-lapse manner over the first and the second vaccination period. After characterizing the basic immune cell profiles, the authors attempted to identify possible factors associated with variable responses to the vaccination among different individuals. They found that baseline expression levels of AP-1 transcription factors, such as FOS and ATF3, are inversely correlated with the T cell responses. To further characterize the factors associated with different induction of these transcription factors, they analyzed gut microbes of the participants. The fucose/rhamnose degradation pathway appeared to be correlated with the inductions of the transcription factors. Overall, I think this manuscript present relevant novel insights based on

substantial amount of original data, revealing the interaction of blood cell profiles and the gut microbes. However, I'd like to point out that some technical concerns should be addressed, mainly for the analysis of gut microbes. Some improvements are also needed on the writing style of the manuscript as described below.

Major Comments:

1. To my knowledge, most of the contents described in the first half of the paper are merely confirmatory to or as expected from the previous studies (for example, <https://www.nature.com/articles/s41586-021-03738-2>; please enrich the reference with the review papers therein). Nevertheless, I consider the main results on the association of AP-1 transcription factors and gut microbes with the high vaccine responses should be unique and interesting. To put more focus on the latter part, the writing style should be revised so that the first part should be shortened. Particularly, the current figures should be more concise to rather enrich the results related to the latter part.

2. Single cell analysis was conducted using the PBMCs stimulated ex vivo. I wonder the same reaction should really occur in vivo, too? Is not it possible to conduct the similar analysis by recruiting the participants showing an appropriate profile after the vaccination?

3. Line 277-: As for the gut microbe analysis, the authors employ PICRUSt2, which is a tool to infer the functional composition based on the 16S rRNA amplicon data. Therefore, changes of the glucose/rhamnose degradation pathway were not directly analyzed. In addition, it is known that other pathways are also involved in the production of SCFA, remaining the possibility that the fructose/rhamnose degradation pathway might not be responsible. Further careful validation analyses should be needed for this issue.

4. Re-analysis of the immune profiles should be needed, having observed the indicated responses of the gut microbes. If the suggested hypothesis is correct, it may make some influence on the Treg-related pathways, which should appear in the former part of the analysis. At least, the authors need to re-analyze the changes in the expression patterns for the genes downstream to the SCFA induction, including the COX2 gene.

Minor Comments:

5. Gut microbes: As this topic is the most important issue, the technical details on the gut microbe analysis should be more clearly explained. The timing of the sampling and the procedure to isolate the genomic DNA, including fecal storage condition, should be described in more details. The parameters and versions of the bioinformatics tools used for the data process should be also specified. Statistics of the sequencing should be also presented somewhere.

6. "Study design" and Line 454- 455: The authors seem to mention that they loaded cells targeting 20,000 cells on 10x Genomics Chromium Controller for their scRNA-seq. It seems that this number is that of "loaded" cells, not "target" cells. In either way. To avoid confusions, please modify the description. Also, please double check about the doublet rates and other traces of technical errors at the same time with presenting the general sequencing stats.

7, Line 87-: In this study, the samples were collected at/before the vaccination on; day 2±1 (first vaccination), day2±1 (second vaccination), day8±1 (second vaccination) and day41±3 (second vaccination). Is there any specific reason to collect samples at these time points? Also, please explain why the indicated time points were selected for the respective assays.

8. Line 103, Figs 2a and 2b: Please specify the precise definition of "seronegative/seropositive for SARS-CoV-2" and provide the reason for the cut-off. Also, for the originally seropositive participants, it is preferable to examine the presence of N-protein antibodies to see possible previous infections of SARS-CoV-2.

9. Line 111 and Line 354: As for the cross-reactivity of T cell, please provide some more detailed information about the previous knowledge, the homology and the used peptides and so on,

somewhere in Supplementary Fig. 2.

10. Line 178: The authors employed qRT-PCR to compare the expression level of IFNB1 between high/low T-cell responders. To what extent were the results consistent with the RNA seq?

11. Line 214-, Fig1: If there are any cases where the stool samples were collected at multiple time points, it is interesting to see the dynamic changes of the gut microbes and the possible correlation with the degree of the side effects.

12. Line 326: In this study, the authors recruited Japanese healthy participants. Would the authors think the results about the gut microbiota should be common to other ethnic groups?

13. Line 370-: As for the CyTOF analysis, the authors showed cell annotation based on its datasets. For the sake of the reproducibility, the authors should provide the list of antibodies used for the current study. If the standard antibody set of the Fluidigm was used, please specify so.

Responses to reviewers' comments.

We greatly appreciate the constructive reviews provided by the reviewers. Considering all of their comments and suggestions, we have performed additional experiments, revised the manuscript text, and provided some new figures. Below, please find our responses to the reviewers' questions and comments. All changes have been marked in red in the revised manuscript and indicated with line numbers in our response. We are confident that our paper has been considerably improved by the reviewers' critiques.

Reviewers' comments:

Reviewer #1 (Remarks to the Author):

Hirota et al. employ a systems biological approach to determine correlates of inter-individuals variation observed in BNT162b2 vaccinees. Using a cohort of healthy volunteers (n=96 participants, of which 86 are seronegative for SARS-CoV-2 at baseline). Using CyTOF, they show that CD14+ monocyte frequency measured 42 days post second dose positively correlates with vaccine-induced T cell responses. Using transcriptomics analysis, they also show that an AP-1 transcription factor expression correlates negatively with T cell responses while a type I interferon response signature positively correlates with T cells. Furthermore, they show that Fucose/rhamnose degradation pathway, measured by metagenomic sequencing, correlates with AP-1 transcription factors thus negatively correlating with the T cell responses. While the objective of the study is of interest to the field, the study needs to be improved significantly.

Thank you very much for the positive comments and insightful suggestions. Considering your suggestions, we performed additional experiments. Particularly, analyses of split cohorts for discovery and validation strengthened our conclusion.

1. A major issue is lack of validation. Since the primary goal of the study is to determine the correlates of variations in vaccine-induced responses, it is critical that the determinates are validated in an independent dataset. I realize they may not have an independent population, but they could randomly distribute the participants (~40 each) into two cohorts, and use one as discovery and the other as validation.

As the reviewer suggested, we analyzed the data of CyTOF, bulk RNA-seq, and 16S rRNA seq in randomly distributed discovery and validation cohorts (n = 43 each). Analysis of the discovery cohort revealed that (i) the monocyte frequency at T5 was correlated with vaccine-induced T cell responses, and (ii) the AP-1 transcription factor module was negatively associated with vaccine-induced T-cell responses. These findings were confirmed in analyzing the validation cohort. However, unlike the analysis of the entire cohort (n = 86), analysis of the discovery cohort did not show a statistically significant correlation between vaccine-induced T cell responses and (i) gut microbial fucose rhamnose degradation, (ii) baseline expression of FOS, FOSB, and ATF3, or (iii) *ex vivo* BNT162b2 RNA-induced IFN responses.

While we agree with the importance of validating findings, the small size of the discovery cohort likely limits chances of discovery. Therefore, we think that it is worthwhile to report discoveries in the analysis of the entire cohort, even though they have not been validated in an independent cohort. In

the revised manuscript, we first described results validated in the discovery/validate cohort analysis, followed by results in the entire cohort analysis. We also mentioned the lack of validation for some data in the Discussion.

[Changes]

1. The Results section has been substantially revised by adding the content related to the discovery/validation cohorts (lines 98-105, 110-121, 128-136, 202-205).
2. The way to randomly select discovery/validation cohorts has been added to the Methods (lines 345-348).
3. The lack of validation of some of our findings has been mentioned as the limitation of the study in the Discussion section (lines 314-316).
4. Figs 1b, 2a, and 3a and Supplementary Figs. 4b, c, and 5c have been added.

2. How were the data normalized for age and sex Figures 3b, c and other datasets?

To remove the effects of age, gender, and fecal sampling timing in partial correlation analysis, we calculated Spearman's correlation coefficient using residuals from linear regression of ranked variables.

[Changes]

In lines 595-597, "For partial correlation tests, we removed the effects of age, gender, and fecal sampling timing from each dataset." was replaced with the above sentence.

3. It is intriguing that the CD14+ monocyte frequencies increase significantly at T5 (Fig. 3e, top panel), i.e., 42 days post second dose while it is relatively stable at the rest of the time points analyzed. What do the authors' think is the reason for such an increase at such a late time point?

Since vaccine-induced inflammatory responses are usually terminated within a week, changes in the frequency of monocytes at 42 days after the second dose might be due to vaccine-induced innate immune memory or trained immunity. Monocytes can maintain altered transcriptional and epigenetic profiles, even after resolution of inflammatory responses induced by BCG, flu, and HIV vaccines and can mediate innate immune memory responses to vaccine targets and unrelated pathogens (Wimmers et al., *Cell*, 2021; Arts et al., *Cell Host Microb.*, 2018; Vaccari et al., *Nat. Med.*, 2018). BCG vaccination enhances myelopoiesis, while decreasing lymphopoiesis, in the bone marrow through persistent epigenetic modification of hematopoietic stem cells, which is plausibly important for innate immune memory of short-lived monocytes (Kaufmann et al., *Cell*, 2018). These epigenetic mechanisms in monocytes or hematopoietic stem cells may be involved in changes in the frequency of monocytes that we observed at T5 after BNT162b2-induced inflammatory responses.

[Changes]

1. The above content has been reflected to the Discussion (lines 307-312).
2. A reference (Kaufmann et al., *Cell*, 2018) has been added.

4. If the monocytes increase in frequency, bulk transcriptomics should also show an increase in monocyte-associated signatures increasing at the same time point (T5). Why did the authors not look at it? This could also be an additional layer of confirmation of the monocyte correlation with T cells?

As the reviewer suggested, we performed an additional bulk RNA-seq analysis of PBMCs collected from high and low T-cell responders in the entire cohort at T5. Consistent with the correlation between the monocyte frequency at this time point and vaccine-induced T cell responses, gene set enrichment analysis showed that a BTM related to monocyte biology was positively associated with T cell responses (Supplementary Fig. 5f).

[Changes]

1. The above content has been added to the Result (lines 150-153).
2. Supplementary Fig. 5f has been added.

5. What do the authors posit as a biological explanation for a negative correlation between frequency of T cell subsets (Fig. 3c) and antigen-specific T cell response?

As mentioned above, we speculate that epigenetic mechanisms in monocytes or hematopoietic stem cells may be involved in changes in the frequency of monocytes that we observed at T5 after BNT162b2-induced inflammatory responses. Further studies are needed to address this possibility and the immunological impact of this change.

[Changes]

1. The above content has been reflected to the Discussion (lines 307-312).
2. A reference (Kaufmann et al., Cell, 2018) has been added.

6. Single-cell RNAseq of PBMCs is a nice way of validating bulk transcriptomics results, and provide further insights into the cellular origin of AP-1 genes. However, it appears that the authors analyzed only baseline PBMCs (stimulated in vitro with mRNA, see next point) and not T4, which is when the bulk transcriptomics analysis showed a correlation. Why is this? I am unclear if the single-cell data confirms AP-1 correlation with T cells.

We are sorry that in the original manuscript, it was not clear that differential expression of AP-1 between high and low T cell responders was detected only at T1 (baseline), but not T4. Given the correlation between baseline AP-1 expression and vaccine-induced T cell responses, we sought to understand the role of baseline AP-1 in various immune cells stimulated with mRNA *ex vivo* using scRNA-seq analysis.

[Changes]

We have specified that AP-1 expression at T1, but not at T4, is negatively associated with T cell responses (lines 132, 141).

7. The in vitro system where they transfect in vitro transcribed mRNA is not comparable with in vivo administration of BNT162b2. The lipid nanoparticle used in BNT162b2 has enormous effects on the innate immune system (see Alameh et al. Immunity 2021). The in vitro mRNA transfection does not have this component.

We agreed and analyzed *IFNBI* expression in PBMCs stimulated with BNT162b2 mRNA encapsulated with lipid nanoparticles (LNP). This showed that LNP-mRNA-induced *IFNBI* expression *ex vivo* was also higher in high-T-cell responders (Supplementary Fig. 6c).

[Changes]

1. The above content has been added (lines 168-170)
2. Supplementary Fig. 6c has been added.

8. The gut microbiome data is interesting but it appears that the stool samples were not synchronized in collection time point. How is that divergence normalized? Overall, the goal of the study is interesting and important, but the manuscript needs significant improvement and importantly, provide validation.

To remove the effects of age, gender, and fecal sampling timing in partial correlation analysis, we calculated Spearman's correlation coefficient using residuals from linear regression of ranked variables.

[Changes]

In line 595-597, "For partial correlation tests, we removed the effects of age, gender, and fecal sampling timing from each dataset." was replaced with the above sentence.

Reviewer #2 (Remarks to the Author):

General

In this manuscript, Hirota et al describe multi-omics analysis of the individuals who received the vaccination of SARS-CoV-2. The analysis covers the ELISpot analysis for the T cells reactive to the spike protein, ELISA assays of the antibodies and CyTOF analysis of PBMC. The sampling was performed in a time-lapse manner over the first and the second vaccination period. After characterizing the basic immune cell profiles, the authors attempted to identify possible factors associated with variable responses to the vaccination among different individuals. They found that baseline expression levels of AP-1 transcription factors, such as FOS and ATF3, are inversely correlated with the T cell responses. To further characterize the factors associated with different induction of these transcription factors, they analyzed gut microbes of the participants. The fucose/rhamnose degradation pathway appeared to be correlated with the inductions of the transcription factors. Overall, I think this manuscript presents relevant novel insights based on a substantial amount of original data, revealing the interaction of blood cell profiles and the gut microbes. However, I'd like to point out that some technical concerns should be addressed, mainly for the analysis of gut microbes. Some improvements are also needed on the writing style of the manuscript as described below.

Thank you very much for the positive comments and valuable suggestions. The manuscript has been greatly improved by conducting some additional experiments and changing the structure of the Results section as you suggested.

Major Comments:

1. To my knowledge, most of the contents described in the first half of the paper are merely confirmatory to or as expected from the previous studies (for example, <https://www.nature.com/articles/s41586-021-03738-2>; please enrich the reference with the review papers therein). Nevertheless, I consider the main results on the association of AP-1 transcription factors and gut microbes with the high vaccine responses should be unique and interesting. To put

more focus on the latter part, the writing style should be revised so that the first part should be shortened. Particularly, the current figures should be more concise to rather enrich the results related to the latter part.

Thank you for the valuable comment. We substantially revised the structure of the Results section to focus on our original findings. We have also added references, including review articles that reported an adaptive immune response to the COVID-19 vaccine (Sette & Crotty, *Immunol Rev.*, 2022; Bai et al., *Front Immunol.*, 2022; Bayart et al., *Microorganisms.*, 2021; Jo et al., *Front Aging.*, 2021; Loyal et al., *Science*, 2021; Woldemeskel. *et al.*, *J Clin Invest.*, 2021).

[Changes]

1. Description of the results has been made more concise (lines 82-95, 110-121).
2. Above references have been added.
3. Previous Fig. 2 (Inter-individual variations in BNT162b2-induced adaptive immunity in our cohort) was moved to Supplementary Figs. 2 and 3.
4. In the new Fig. 2 (CyTOF data), only our original finding (association between late monocyte response and T-cell response) has been shown. Other data were moved to Supplementary Fig. 4.

2. Single cell analysis was conducted using the PBMCs stimulated ex vivo. I wonder the same reaction should really occur in vivo, too? Is not it possible to conduct the similar analysis by recruiting the participants showing an appropriate profile after the vaccination?

Due to the limited research budget and difficulty in recruitment of subjects for assessment of the association between baseline parameters and vaccine responses, we gave up recruiting a new cohort to perform another scRNA-seq analysis. To address the reviewer's question, we analyzed expression of genes related to IFN response (*IFIT1*, *IFNB1*, *IRF7*, *PSLCR1*) and early T-cell response (*CCL5*, *GZLY*, *NKG7*) at T2 by qPCR (n = 86). As shown in the following figure, there was no statistically significant correlation between expression of these genes at T2 and T-cell responses.

It should be noted that the sampling timing after mRNA stimulation differs between ex vivo assay and

PBMC collection at T2 (6-16 h in ex vivo assay vs day 2 ± 1 in T2 sampling). In addition, as described in the manuscript, relatively large time lags in our blood sampling did not allow us to evaluate dynamic gene expression in BNT162b2-induced innate immunity (lines 158-160). Thus, we consider that our T2 data analysis is not conclusive and should not be included in the manuscript.

[Changes]

We mentioned that in vivo relevance of our ex vivo finding has not been demonstrated in the paragraph of limitation of this study in the Discussion (lines 320-321).

3. Line 277-: As for the gut microbe analysis, the authors employ PICRUST2, which is a tool to infer the functional composition based on the 16S rRNA amplicon data. Therefore, changes of the fucose/rhamnose degradation pathway were not directly analyzed. In addition, it is known that other pathways are also involved in the production of SCFA, remaining the possibility that the fructose/rhamnose degradation pathway might not be responsible. Further careful validation analyses should be needed for this issue.

To evaluate gut microbial fucose degradation activity, we anaerobically incubated stool slurry in the presence of fucose *in vitro* for 24 h and measured fucose levels in the culture media. The results showed that samples predicted by PICRUST2 to have high fucose-rhamnose degrading activity reduced fucose in the medium more than those predicted to have low activity (Fig. 6f), confirming the prediction.

[Changes]

1. This above content has been added to the Results (lines 228-232).
2. Fig. 6f has been added.

4. Re-analysis of the immune profiles should be needed, having observed the indicated responses of the gut microbes. If the suggested hypothesis is correct, it may make some influence on the Treg-related pathways, which should appear in the former part of the analysis. At least, the authors need to re-analyze the changes in the expression patterns for the genes downstream to the SCFA induction, including the COX2 gene.

As the reviewer suggested, we tested whether gut microbial fucose/rhamnose degradation is associated with *PTGS2* (also known as *COX2*) expression and Treg frequency. The results revealed that fucose/rhamnose degradation was correlated with *PTGS2* expression in PBMCs, but not with Treg frequency at T1 (Fig. 7c and Supplementary Fig. 10d). *PTGS2* expression was also positively correlated with expression of AP-1 factors, FOS, FOSB, and ATF3, and inversely correlated with vaccine-induced T cell responses (Fig. 7d, e). Taken together, these data support the hypothesis that gut microbial fucose/rhamnose degradation may upregulate *PTGS2*/*PGE2* expression in PBMCs probably through SCFAs, thereby promoting AP-1 expression. Our data suggested that fucose/rhamnose degradation is unlikely to be associated with Tregs in PBMCs.

[Changes]

1. The above content has been added to the Results (lines 244-251) and Discussion (lines 285-287).
2. Fig. 7c-e and Supplementary Fig. 10d have been added.

Minor Comments:

5. Gut microbes: As this topic is the most important issue, the technical details on the gut microbe analysis should be more clearly explained. The timing of the sampling and the procedure to isolate the genomic DNA, including fecal storage condition, should be described in more details. The parameters and versions of the bioinformatics tools used for the data process should be also specified. Statistics of the sequencing should be also presented somewhere.

We added details of these analyses. New Supplementary Fig. 9f shows sampling timing of each subject.

[Changes]

1. The detail of the analyses has been added (lines 227-228).
2. Supplementary Fig. 9f has been added.

6. “Study design” and Line 454- 455: The authors seem to mention that they loaded cells targeting 20,000 cells on 10x Genomics Chromium Controller for their scRNA-seq. It seems that this number is that of “loaded” cells, not “target” cells. In either way. To avoid confusions, please modify the description. Also, please double check about the doublet rates and other traces of technical errors at the same time with presenting the general sequencing stats.

Thank you for pointing out our mistake. We have corrected this and included details on the number of loaded cells, captured cells, and the multiplet rate in the new supplementary table.

[Changes]

1. Cell number information has been corrected as follows: “Single-cell suspensions (about 20,000 cells) were then loaded on the 10X Genomics Chromium Controller.” (lines 500-501).
2. A new supplementary table has been added.

7, Line 87-: In this study, the samples were collected at/before the vaccination on; day 2 ± 1 (first vaccination), day 2 ± 1 (second vaccination), day 8 ± 1 (second vaccination) and day 41 ± 3 (second vaccination). Is there any specific reason to collect samples at these time points? Also, please explain why the indicated time points were selected for the respective assays.

A previous study detected a remarkable increase in expression of innate immune genes, 1 day after the first dose and 1-7 days after the second dose (Arunachalam et al, Nature, 2021). Accordingly, we collected blood samples at multiple time points to evaluate baseline responses (T1: before vaccination: T1), innate immune responses (T2: day 2 ± 1 after the first dose; T3: day 2 ± 1 after the second dose; T4: day 8 ± 1 after the second), and long-term adaptive immune responses (T5: day 41 ± 3 after the second dose).

[Changes]

The above content has been added to the Results (lines 72-79).

8. Line 103, Figs 2a and 2b: Please specify the precise definition of “seronegative/seropositive for SARS-CoV-2” and provide the reason for the cut-off. Also, for the originally seropositive participants, it is preferable to examine the presence of N-protein antibodies to see possible previous infections of SARS-CoV-2.

Thank you for pointing out this ambiguous point. That should have been “seronegative/seropositive for SARS-CoV-2 spike.” Additional anti-N IgG ELISA analysis of subjects seropositive for SARS-CoV-2 spike at T1 (n = 9) revealed that levels of antibody against SARS-CoV-2 S and N were correlated at T1 (Supplementary Fig. 2c).

[Changes]

1. “seropositive/seronegative for SARS-CoV-2” has been corrected to “seropositive/seronegative for SARS-CoV-2 spike” in the Results (lines 85-88, 100, 126-127).

2. Supplementary Fig. 2c has been added.

9. Line 111 and Line 354: As for the cross-reactivity of T cell, please provide some more detailed information about the previous knowledge, the homology and the used peptides and so on, somewhere in Supplementary Fig. 2.

We have added the following information: Each peptide pool consists of 15-mer peptides overlapping 11 amino acids, covering the entire spike protein from SARS-CoV-2 (UniProt ID: P0DTC2), HCoV-229E (UniProt ID: P15423), HCoV-NL63 (UniProt ID: Q6Q1S2), HCoV-HKU1 (UniProt ID: Q5MQD0), and HCoV-OC43 (UniProt ID: P36334). As shown in Supplementary Fig. 2, SARS-CoV-2 showed 27-31% identity in amino acid (aa) sequences of spike with the four HCoVs analyzed. There were high levels of identity (about 60%) in aa sequences of spike between HCoV-229E and HCoV-NL63, HCoV-HKU1 vs HCoV-OC43.

[Changes]

This content has been added to the Methods (lines 386-389).

10. Line 178: The authors employed qRT-PCR to compare the expression level of *IFNB1* between high/low T-cell responders. To what extent were the results consistent with the RNA seq?

Our bulk-RNA-seq analysis also showed that *IFNB1* expression was correlated with vaccine-induced T-cell responses (Fig. 4c).

[Changes]

1. This content has been added to the Results (lines 164-165).

2. Fig. 4c has been added.

11. Line 214-, Fig1: If there are any cases where the stool samples were collected at multiple time points, it is interesting to see the dynamic changes of the gut microbes and the possible correlation with the degree of the side effects.

Because we did not collect stool samples at multiple time points, we cannot address this question with our own data. However, a previous study reported metagenomic data from the gut microbiome before and after COVID-19 mRNA vaccination (Ng et al., Gut, 2022). That study detected changes in microbiome composition one month after COVID-19 mRNA vaccination and found that several bacterial species were negatively correlated with vaccine-induced adverse effects. Using their data, we examined whether the vaccine affects abundance of gut microbial fucose/rhamnose degradation.

There was no significant change in the abundance of fucose/rhamnose degradation 1 month after vaccination (Supplementary Fig. 9e), suggesting that the activity of this metabolic pathway is largely not influenced by COVID-19 mRNA vaccination.

[Changes]

1. The content related to the effect of COVID-19 vaccination on fucose/rhamnose degradation has been added to the Results (lines 223-227).
2. Supplementary Fig. 9e has been added.

12. Line 326: In this study, the authors recruited Japanese healthy participants. Would the authors think the results about the gut microbiota should be common to other ethnic groups?

Because genes related to fucose/rhamnose degradation are encoded by genomes of gut bacterial species that are commonly detected in diverse ethnic groups, we speculate that inter-individual variation in this metabolic activity and its association with COVID-19 vaccine-induced T cell responses are not limited to the Japanese population. Whether our findings are relevant to other ethnic groups is an important issue for future studies.

[Changes]

The above content has been added to the Discussion (line 320-321).

13. Line 370-: As for the CyTOF analysis, the authors showed cell annotation based on its datasets. For the sake of the reproducibility, the authors should provide the list of antibodies used for the current study. If the standard antibody set of the Fluidigm was used, please specify so.

We have added the antibody information.

[Changes]

The antibody information has been added to the Method (lines 407-413).

REVIEWERS' COMMENTS:

Reviewer #1 (Remarks to the Author):

I appreciate that the authors have made considerable efforts to address all my questions. The discovery-validation analysis clearly indicates some features as more reliable than others. Although I am not convinced that "trained immunity" explains the observed increase in monocyte frequency on day 42, it's good that this point has been added to the discussion section. Overall, the authors have responded to my comments satisfactorily.

Reviewer #2 (Remarks to the Author):

First of all, I appreciate the substantial efforts of the authors to revise this manuscript. Thanks to their extensive analysis and detailed discussion, I believe the manuscript has been very much improved. All the concerns, especially those from technical viewpoints, have been almost completely addressed. I sincerely hope the authors continue their efforts to further clarify how the human body, as a whole, react to the infectious disease, especially for a larger number of individuals.

Responses to reviewers' comments.

Reviewer #1 (Remarks to the Author):

I appreciate that the authors have made considerable efforts to address all my questions. The discovery-validation analysis clearly indicates some features as more reliable than others. Although I am not convinced that "trained immunity" explains the observed increase in monocyte frequency on day 42, it's good that this point has been added to the discussion section. Overall, the authors have responded to my comments satisfactorily.

We agree that the discovery-validation analysis greatly improved our manuscript. Thank you very much for your constructive review.

Reviewer #2 (Remarks to the Author):

First of all, I appreciate the substantial efforts of the authors to revise this manuscript. Thanks to their extensive analysis and detailed discussion, I believe the manuscript has been very much improved. All the concerns, especially those from technical viewpoints, have been almost completely addressed. I sincerely hope the authors continue their efforts to further clarify how the human body, as a whole, react to the infectious disease, especially for a larger

number of individuals.

We greatly appreciate your constructive suggestions. In the revision of the manuscript, we learned many important points in the study of commensal microbiota. Thank you very much.